

# Inorganic Carbon and Water Masses in the Irminger Sea since 1991

Friederike Fröb[1,2], Are Olsen[1,2], Fiz F. Pérez[3], Maribel I. García-Ibáñez[4], Emil Jeansson[4], Abdirahman Omar[4], and Siv K. Lauvset[4]

[1]Geophysical Institute, University of Bergen, Bergen 5007, Norway.
[2]Bjerknes Centre for Climate Research, Bergen 5007, Norway.
[3]Instituto de Investigaciones Marinas (IIM-CSIC), Vigo 36208, Spain.
[4]Uni Research Climate, Bjerknes Centre for Climate Research, Bergen 5008, Norway.

*Correspondence to:* Friederike Fröb (friederike.frob@uib.no) and Are Olsen (are.olsen@uib.no)

**Abstract.**

The subpolar gyre region in the North Atlantic is a major sink for anthropogenic carbon. While the storage rates show large interannual variability related to atmospheric forcing, less is known about variability in the natural Dissolved Inorganic Carbon (DIC) and the combined impact of variations in the two components on the total DIC inventories. Here, data from 15 cruises in the Irminger Sea covering 1991-2015 were used to determine changes in total DIC and its natural and anthropogenic components in relation to the distribution and evolution of the main water masses. The inventory of DIC increased by $1.43\pm0.17$ mol m$^{-2}$yr$^{-1}$ over the period, mainly driven by the increase in anthropogenic carbon ($1.84\pm0.16$ mol m$^{-2}$yr$^{-1}$), but partially offset by a loss of natural DIC ($-0.57\pm0.22$ mol m$^{-2}$yr$^{-1}$). Changes in the carbon storage rate can be driven by concentration changes in the water column, for example due to ageing of water masses, or by changes in the distribution of water masses with different concentrations, either by local formation or advection. A decomposition of the trends into their main drivers showed that variations of natural DIC inventories are mainly driven by changes in the layer thickness of the main water masses, while anthropogenic carbon is most affected by concentration changes. The storage rates of anthropogenic carbon are sensitive to data selection, while changes in DIC inventory show a robust signal on short timescales, associated with the strength of convection.

## 1 Introduction

Since the industrial revolution, atmospheric $CO_2$ levels have been increasing almost exponentially as a result of human activities such as fossil fuel burning, cement production, and land use changes. The global ocean has acted as strong sink for this anthropogenic $CO_2$ (Sabine et al., 2004) and is currently taking up approximately 25 % of the annual emissions (Le Quéré et al., 2016). While the ocean has capacity to store almost all of the anthropogenic $CO_2$ released to the atmosphere, the emissions currently outpace the oceanic absorption rates (Sabine and Tanhua, 2010). This is because the transport of anthropogenic $CO_2$ from the atmosphere into the ocean interior is limited by the rate of vertical exchange between the surface and deep ocean (Sarmiento and Gruber, 2002). Warming of the ocean will decrease this rate as a consequence of the increased stratification, and Earth System Models predict a decline in oceanic anthropogenic $CO_2$ uptake efficiency over the 21st century (Friedlingstein et al., 2006; Schwinger et al., 2014). Warming of the ocean will also affect $CO_2$ solubility, primary production





and other factors governing the distribution and inventory of natural carbon in the ocean (Arora et al., 2013; Schwinger et al., 2014). It is important to constrain the magnitude of these feedbacks for policy planning, but current estimates vary significantly among models. Observational based quantitative and qualitative insight in carbon cycle climate interactions are important for the further improvement of projections of the future ocean carbon cycle.

Over the past three decades, ocean $CO_2$ chemistry data have been collected on a regular basis in the world's oceans. As the observational record extends, direct evidence of the climate sensitivity of the marine carbon cycle emerges. For example, the Southern Ocean carbon sink exhibits clear variations in response to atmospheric circulation patterns; the sink was weakening from the early 1980s to the early 2000s (Le Quéré et al., 2007), but has strengthened in the more recent decades

(Landschützer et al., 2015). In the subarctic western North Pacific, measurements from 1992 to 2008 at the two time series stations KNOT and K2 reveal decadal trends in total dissolved inorganic carbon (DIC), related to alkalinity driven reductions in $CO_2$ outgassing (Wakita et al., 2010). In the Mediterranean Sea, changes in the large-scale circulation result in variability of the anthropogenic $CO_2$ concentration (Touratier and Goyet, 2009). Within the subpolar North Atlantic (SPNA), high-quality carbon data have been collected almost every second year since the early 1990s (Olsen et al., 2016), enabling the determination

of subdecadal variability. This shows relationships between the anthropogenic $CO_2$ storage rate and the extent and intensity of ventilation processes, primarily driven by the North Atlantic Oscillation (NAO) (Pérez et al., 2010; Wanninkhof et al., 2010; Fröb et al., 2016; Woosley et al., 2016).

The SPNA is a key region for the storage and transport of $CO_2$ in the global ocean (Sabine et al., 2004). While a response of

anthropogenic $CO_2$ storage to atmospheric forcing has been determined as mentioned above, less is known about variations in natural DIC, any relations to atmospheric forcing and relevance for total DIC inventories (Tanhua and Keeling, 2012). Here, we analyse changes in total DIC and its natural and anthropogenic components in the central SPNA, the Irminger Sea, in relation to the distribution and evolution of water masses over a 24-year period from 1991 to 2015, covering three periods of variable convective activity (Fröb et al., 2016).

**2 Hydrographic setting**

The Irminger Sea, a central sea in the SPNA (Fig. 1), is a climatically sensitive area with strong hydrographic contrasts. The SPNA circulation pattern has been extensively presented in the literature; here the description follows Lavender et al. (2005) and Våge et al. (2011). In the upper ocean, the East Greenland Current (EGC) carries cold and fresh water of Arctic origin southwards, in the west, close to the shelf of Greenland. In the east, the Irminger Current (IC) carries warm and salty water

northwards along the Reykjanes Ridge. The salinity and temperature signature of these warm water masses is affected by the strength and shape of the subpolar gyre (SPG) (Häkkinen and Rhines , 2004; Hátún et al., 2005). South of the Denmark Strait, they largely recirculate to the south. In the centre of the cyclonic circulation of the Irminger Gyre, preconditioning conditions for convection are fulfilled (Marshall and Schott, 1999; Bacon et al., 2003; de Jong et al., 2012) and depending on heat loss,





deep convection can occur (Pickart et al., 2003a; Våge et al., 2008, 2011; Fröb et al., 2016; de Jong and de Steur, 2016). The extent and strength of convective processes are mainly driven by the state of the NAO, which is the leading mode of atmospheric variability over the mid-North Atlantic (Curry et al., 1998; Hurrell and Deser, 2009).

5  At depth, the circulation in the Irminger Sea is mainly characterized by the Denmark Strait Overflow Water (DSOW) and the Iceland-Scotland Overflow Water (ISOW). DSOW, to the west, is a relatively recently ventilated water mass enriched in oxygen ($O_2$) as well as other dissolved atmospheric gases. It is composed of several water masses originating from the Arctic Ocean and the Nordic Seas (Tanhua et al., 2005; Jeansson et al., 2008). ISOW originates from intermediate waters of the Nordic Seas, which are modified as they flow through the Iceland Basin to the Irminger Sea (Hansen and Østerhus, 2000). In combination

10 with ISOW and DSOW, Labrador Sea Water (LSW) forms North Atlantic Deep Water (NADW) (Dickson and Brown, 1994), the key component of the lower limb of the Atlantic Meridional Overturning Circulation. Two main LSW classes are identified in the SPNA: the $LSW_{1987-1994}$ and the $LSW_{2000}$. $LSW_{1987-1994}$, from now on called classical LSW (cLSW), is a dense, cold and relatively fresh water mass with high concentrations of dissolved atmospheric gases, formed by the recurring winter convection in the mid-1980s and mid-1990s in the Labrador and Irminger Seas (Lazier et al., 2002; Pickart et al., 2003b;

15 Yashayaev et al., 2007). After 2000, the lighter $LSW_{2000}$ or upper Labrador Sea Water (uLSW) has largely replaced cLSW (Yashayaev et al., 2007).

## 3 Data

Data from 15 cruises in the Irminger Sea covering 1991-2015 are used in this study (see Table 1). Data from the first 13 cruises were extracted from the GLODAPv2 data product, which provides bias-corrected, cruise-based, interior ocean data (Key et al.,

20 2015; Olsen et al., 2016). The more recent data are from the 2012 OVIDE cruise (expocode: 29AH20120623) and the 2015 SNACS cruise (expocode: 58GS20150410). In order to minimize seasonal bias due to primary production, the upper 100 m of the water column are excluded from the inventory analysis. The region between 40.5° W and 31.5° W was covered by all 15 cruises.

25  All cruises intersect the ocean currents of the Irminger Sea described in the previous section. The cruises occupied either WOCE section A01/AR07E, the FOUREX or the OVIDE section, and locations are presented in Fig. 1. In order for the sections to be fully comparable, a coordinate transformation was performed for the 1997 FOUREX data. The latitude and longitude coordinates were rotated to the AR07E section using Cape Farewell as pivot point. Adjusting the distance between the stations ensured that the adjusted coordinates of the station over the Reykjanes Ridge on the FOUREX line matched the station over the

30 Reykjanes Ridge on the AR07E line. The inventory estimates are sensitive to depth, therefore the pressure coordinates of all cruises were normalized. The location of every station for all cruises was mapped to the one arc-minute global relief model of the Earth's surface (Amante and Eakins, 2009) using a nearest-neighbour interpolation. The ratio between the bottom depth of this bathymetry and the reported cruise station bottom depth was multiplied with the pressure coordinates of each station. This





normalization step mainly affected the adjusted FOUREX data, while for the other cruises the normalization changed sampling depths by less than 20 meters.

The accuracy of the GLODAPv2 data product is better than 0.005 in salinity, 1 % in $O_2$, 2 % in nitrate ($NO_3$), 2 % in
silicate ($SiO_2$), 2 % in phosphate ($PO_4$), 4 $\mu$mol kg$^{-1}$ in DIC and 6 $\mu$mol kg$^{-1}$ in total alkalinity ($A_T$) (Olsen et al., 2016). For 29AH20120623, the overall accuracy of $NO_3$, $PO_4$ and $SiO_2$ was 1 %; the accuracy of DIC was 2 $\mu$mol kg$^{-1}$ and for $A_T$ it was 4 $\mu$mol kg$^{-1}$ (Ríos et al., 2015; García-Ibáñez et al., 2016). For the SNACS cruise in 2015, pressure, conductivity, temperature and dissolved $O_2$ were directly measured with a Seabird 911+ CTD profiler. At every station, water samples were obtained at 12 depths using Niskin bottles, and used to calibrate the CTD measurements following the Global Ocean
Ship-based Hydrographic Investigations Program (GOSHIP) calibration procedure (Hood et al., 2010). The accuracy of bottle salinities, analysed with a salinometer, was ±0.003. The accuracy of $O_2$ concentration measured with Winkler titration using a potassium iodate solution as a standard was 0.2 $\mu$mol kg$^{-1}$. The precision was better than 2 % in $PO_4$, 1 % in $SiO_2$ and 1 % in $NO_3$ as evaluated using samples drawn from sets of Niskin bottles tripped at the same depth. DIC and $A_T$ was measured according to Dickson et al. (2007), with an accuracy of 2 $\mu$mol kg$^{-1}$ for both (Fröb et al., 2016).

The seawater $CO_2$ chemistry can be fully described if at least two of the four variables DIC, $A_T$, $CO_2$ partial pressure or pH are known. The measured variables at each of the 15 cruises are listed in Table 1. For six cruises, $A_T$ and pH were measured, therefore DIC was calculated for these, using the dissociation constants of Lueker et al. (2000). For three cruises, only DIC was measured. For these, $A_T$ was approximated using the salinity-alkalinity relationship for the North Atlantic of Lee et al. (2006).
This relationship is defined for the surface ocean only, therefore its validity for the deep Irminger Sea was tested (Appendix A). The mean difference between approximated and the measured $A_T$ data available was less than 5 $\mu$mol kg$^{-1}$; this is better than the target accuracy of $A_T$ of the GLODAPv2 data product. No bias with depth or position was evident.

## 4  Method

The total DIC concentration is partitioned into its natural and anthropogenic components (DIC = $DIC_{nat}$ + $C_{ant}$). The $C_{ant}$
concentration was estimated with the $\varphi C_T^\circ$ method (see section 4.1). The $DIC_{nat}$ concentration is the difference between DIC and $C_{ant}$. For all cruises, the column inventories were estimated for DIC, $DIC_{nat}$ and $C_{ant}$. The inventories are sensitive to depth, therefore column inventories were only estimated for the part of the transect covered by all 15 cruises, between 40.5° W and 31.5° W. The column inventory is the concentration profile integrated over the entire water column (Tanhua and Keeling, 2012):

$$Inv_s = \int_0^z c_s * \varrho \, dz \tag{1}$$





Here, $Inv_s$ is the column inventory of any species $s$, $c_s$ its concentration, $\varrho$ the density at in situ temperature and pressure and $z$ the depth of the water column. The storage rate is the slope of a linear least-squares regression over the mean column inventories with time. The standard error of the slope is the error of the storage rate. Changes in inventories can be caused by changes in the distribution of water masses with different species concentrations or by changes in species concentration within

the water masses. The distribution of water masses was determined using an extended Optimum MultiParameter analysis (eOMP, see section 4.2). The change in concentration within each water mass was determined by applying the concept of water mass mixing averaged concentration, i.e. archetypal concentration (Álvarez-Salgado et al., 2013) (see section 4.2). Assuming linearity, the inventory changes can then be decomposed into contributions from changes in the archetypal concentration of the source water types (SWTs) and from changes in layer thickness of each water mass:

$$\frac{dInv_{tot}}{dt} = \sum_{WM} \left( \frac{\partial Inv}{\partial c}\frac{dc}{dt} + \frac{\partial Inv}{\partial z}\frac{dz}{dt} \right) \qquad (2)$$

Here, $\frac{\partial Inv}{\partial c}\frac{dc}{dt}$ is the mean layer thicknesses with variable archetypal SWT concentrations, while $\frac{\partial Inv}{\partial z}\frac{dz}{dt}$ can be calculated as the mean archetypal SWT concentrations multiplied by the layer thickness changes over a specific time period. Hence, the two drivers for the observed inventory variability of total DIC and its natural and anthropogenic components can be identified.

### 4.1 Anthropogenic CO$_2$ calculation

The $\varphi C_T^\circ$ method was applied to all cruises in the Irminger Sea to estimate C$_{ant}$ concentrations (Pérez et al., 2008; Vázquez-Rodríguez et al., 2009, 2012). The $\varphi C_T^\circ$ method is a back-calculation method that follows the same principles as the $\Delta C^*$ method of Gruber et al. (1996). In the $\varphi C_T^\circ$ method, Cant is quantified as the difference between the preformed DIC at the time $t$ and at preindustrial times ($\pi$): $C_{ant} = DIC^{\circ,t} - DIC^{\circ,\pi}$. $DIC^{\circ,t}$ is calculated by correcting the measured DIC for changes due to remineralisation of organic matter and CaCO$_3$ dissolution, while $DIC^{\circ,\pi}$ is quantified as the sum of the saturated DIC concentra-

tion with respect to the preindustrial atmosphere and the air-sea CO$_2$ disequilibrium ($\Delta$C$_{dis}$). The major advantage of the $\varphi C_T^\circ$ method over other back-calculation methods is that data for age tracers, such as chlorofluorocarbons (CFCs), are not needed. For A$_T^\circ$ and $\Delta$C$_{dis}$ the parametrizations developed by Vázquez-Rodríguez et al. (2012) were adopted. These were determined using data of the subsurface layer (100-200 m), which represents and preserves conditions during water mass formation (Vázquez-Rodríguez et al., 2012). The $\varphi C_T^\circ$ method also takes any spatial and temporal variability of $\Delta$C$_{dis}$ into

account. Further, the parametrized A$_T^\circ$ is corrected for effects of CaCO$_3$ dissolution changes and the sea surface temperature increase since preindustrial times. Overall, the uncertainty of $\varphi C_T^\circ$ derived C$_{ant}$ has been reported to be 5 $\mu$mol kg$^{-1}$ (Vázquez-Rodríguez et al., 2009).

### 4.2 Extended Optimum MultiParameter Analysis (eOMP)

The Optimum MultiParameter (OMP) analysis (Tomczak and Large, 1989) is used to estimate the contribution of water masses,
which are represented through SWTs, to each water parcel along the Irminger Sea sections. The OMP analysis assumes that all hydrographic parameters describing the water masses are affected by the same mixing processes. For each sampling point



the contribution of the various water masses is quantified from an over-determined system of linear mixing equations, which is solved in a non-negative least square sense:

$$Gx - d = R \qquad (3)$$

where $G$ is the SWT matrix containing their properties, $x$ the relative contributions of each SWT to the sample, $d$ the observed data and $R$ the residual. The OMP was further developed into the extended OMP (eOMP) analysis, which is used here, by Karstensen and Tomczak (1998) and Poole and Tomczak (1999). This accounts for the non-conservative behaviour of $O_2$ and nutrients by using Redfield ratios. In the eOMP, the remineralization of $NO_3$ and $PO_4$ is numerically related to an oxygen consumption rate, which, if multiplied with a pseudo-age, is similar to apparent oxygen utilization (AOU) (Poole and Tomczak, 1999). OMP and eOMP analyses have previously been used to describe in detail the origin, pathways and transformation of the main water masses in the SPNA (Tanhua et al., 2005; Álvarez et al., 2005; García-Ibáñez et al., 2015). Here, the SWT properties were defined based on the cruise data from 1991, assuming that the properties of the SWTs do not significantly change over time. The data for potential temperature ($\theta$), salinity, $O_2$, $NO_3$, $PO_4$, $SiO_2$ and potential vorticity (PV) were used to characterize 11 SWTs that combined encompass the property features in the Irminger Sea (Fig. 2). The properties of the SWTs are provided in Table 2, including their standard deviations. These values were determined from the 10 % of data in the relevant density class that were closest to the property maximum or minimum used to delineate the SWT. For example, if a SWT was defined as a salinity minimum, all data points within a specific potential density ($\sigma$) range were sorted by salinity and the mean and standard deviation over the first 10 % of the data points, gave the salinity properties for that SWT. The approximate locations of all SWTs are shown in Fig. 3.

DSOW is the densest water mass in the Irminger Sea and defined as an $O_2$ maximum at $\sigma_2$ levels denser than $37.10\,\mathrm{kg\,m^{-3}}$ (Yashayaev et al., 2007). ISOW is defined as a salinity maximum between 36.89 and $37.10\,\mathrm{kg\,m^{-3}}$ ($\sigma_2$) and $\theta$ between 2.3 and $2.6°$ C. North East Atlantic Deep Water (NEADW) is formed by the entrainment of ISOW with surrounding waters, mainly deep water of Antarctic origin. In the North Atlantic two classes have been identified, upper and lower NEADW (uNEADW and lNEADW) (Castro et al., 1998), but in the Irminger Sea lNEADW is non-existent (McCartney, 1992), while uNEADW was identified as a maximum in $SiO_2$ below 2500 m. The mid-depth weakly stratified layer of cLSW in the Irminger Sea was identified by a PV and salinity minimum between 36.90 and $36.94\,\mathrm{kg\,m^{-3}}$ ($\sigma_2$). uLSW is less dense than cLSW due to its slightly different $\theta$-salinity signature and was identified as a minimum in PV in the $36.81\text{-}36.87\,\mathrm{kg\,m^{-3}}$ $\sigma_2$ range.

The Icelandic Slope Water (IcSW), the Intermediate Water (IW), the Irminger Sea Water (ISW) and the uLSW are typically found at intermediate depths. IcSW is a warm and saline water mass, close the Reykjanes Ridge on the Iceland Slope (Tanhua et al., 2005; Yashayaev et al., 2007). Here, IcSW was defined by a minimum in $O_2$, occupying the 36.80-36.86 kg $\mathrm{m^{-3}}$ $\sigma_2$ range, effectively separating uLSW and cLSW. IW is a saline water mass, depleted in $O_2$, and of southern origin (Sarafanov et al., 2008). IW was identified by $O_2$ values below 250 $\mu$mol $\mathrm{kg^{-1}}$ at $\sigma_0$ between 27.45 and 27.65 kg $\mathrm{m^{-3}}$. ISW

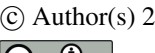



is fresh, elevated in $O_2$, and found between 4 and 5° C.

Finally the Subpolar Mode Water (SPMW), the Subarctic Intermediate Water (SAIW) and the North Atlantic Central Water (NACW) are all typically found in the upper Irminger Sea. SPMW is oxygenated and of subpolar origin. It was defined as a salinity maximum in the 7-8° C $\theta$ range. SAIW is a salinity minimum in the 6.5-7.5° C $\theta$ range. NACW was defined as the salinity maximum for $\theta$ above 9° C.

The seven hydrographic parameters, describing the SWTs, limit the number of SWTs included in one eOMP analysis to a maximum of seven. In addition, the required mass conservation over-determines the system of linear equations. Therefore, the Irminger Sea was divided into four regions and for each data point in each region, an eOMP analysis was carried out with a subset of the 11 SWTs. In the deep ocean ($\sigma_0 \geq 27.76$ kg m$^{-3}$), SWTs were limited to DSOW, ISOW, uNEADW, cLSW and IcSW. In the intermediate ocean ($27.61 \leq \sigma_0 < 27.76$ kg m$^{-3}$), only ISW, IW, IcSW, uLSW and cLSW were included. The upper ocean ($\sigma_0 < 27.61$ kg m$^{-3}$) was split into two basins: east of Reykjanes Ridge (NACW, SPMW, IW and SAIW) and west of the Reykjanes Ridge (ISW, SPMW, IW and SAIW). The data presented in Fig. 2 are classified according to these 'mixing figures'. The equations were normalized and weighted, accounting for differences in measurement accuracies and potential environmental variability. Weights were assigned according to the variability and accuracy of the parameters following García-Ibáñez et al. (2015). The highest weight was assigned to mass to ensure its conservation. The second highest weights were assigned to $\theta$ and salinity, because they are the most accurate. PV was weighted high as well due to its good accuracy and to enable resolution of both LSW classes. The eOMP results in a ratio $r_{ij}$, this describes the contribution of each SWT to each data point in space and time.

For reasons of simplicity, the number of SWTs were reduced from 11 to 9, by performing composite analyses for uNEADW and IcSW. The uNEADW was determined to be a composite of 26 % ISOW, 14 % LSW, 58 % lNEADW and 2 % Mediterranean Water (MW), based on salinity, $\theta$ and $SiO_2$ following (van Aken, 2000). Properties for the SWTs representing lNEADW and MW were taken from García-Ibáñez et al. (2015). A decomposition based on salinity and $\theta$ showed that IcSW is a composite of 30 % ISOW, 20 % cLSW and 50 % IW.

In order to determine the SWT concentrations of DIC, DIC$_{nat}$ and C$_{ant}$, the mixing-weighted concentration or archetypal concentration, $C_i$, of these species was calculated for each SWT, $i$ (Álvarez-Salgado et al., 2013):

$$C_i = \frac{\sum_j r_{ij} \times C_j}{\sum_j r_{ij}} \tag{4}$$



Here, the concentration in each sampling point $j$, $C_j$, is multiplied with the ratio of the SWT in that point $r_{ij}$. When summed over all points and divided by the total fraction the SWT occupies, this estimates $C_i$. Further, the layer thickness, $Th$, of each SWT at each station is estimated according to:

$$Th_{ik} = \frac{\sum_j r_{ij}}{n_k} \times d_{k,max} \tag{5}$$

Here, the sum of the fractions over all sampling point divided by the number of sampling points at each station, $n_k$, and multiplied by its bottom depth, $d_{k,max}$, gives the layer thickness per station for each SWT. The mean over all stations of the Irminger Sea transect is the mean layer thickness.

### 4.3 Uncertainty analysis

Uncertainties for the distribution of water masses result from measurement uncertainties and errors of the eOMP analysis.
Here, the largest source of errors is the definition of the SWTs. The SWT matrix needs to represent the known features of the circulation (Tanhua et al., 2005), but temporal shifts in SWT characteristics cannot be accounted for with the eOMP analysis. A measure of uncertainty is given by the difference between measured and eOMP calculated values, the residual $R$ in Eq. (3). The total residual, calculated by taking the square of the largest parameter residual at each sampling point (García-Ibáñez et al., 2015), and the individual parameter residuals are shown in Fig. 4. Below 1200 m, the total residual is close to zero, as are the
residuals of $\theta$, salinity and $O_2$. In the intermediate and surface ocean these residuals increase, in particular for $O_2$, which might be a consequence of gas exchange. The residuals of $PO_4$, $NO_3$ and $SiO_2$ are larger, as expected due to their lower weights in the eOMP, and do not show any trend with depth. The mean error for each parameter is listed in Table 2. These are of similar magnitude as the errors determined for the Irminger Sea eOMP analysis by Tanhua et al. (2005).

In order to test how robust the results of the eOMP analysis are, a Monte-Carlo simulation was performed (Tanhua et al., 2005). The properties of the SWT matrix were randomly perturbed, within the standard deviation of each parameter. 100 of such perturbed SWT matrices were created and the eOMP was solved for each perturbed system. This allows for quantification of the sensitivity of the eOMP to potential temporal variations of the SWT properties. The standard deviation of the mean SWT contribution over all 100 perturbations is shown in the last column of Table 2. The uncertainties are generally low, hence the
robustness of the eOMP analysis is high.

The layer thickness uncertainties were estimated by scaling the averaged standard deviation of each SWT, which were quantified with the Monte-Carlo simulation, to the width and depth of the Irminger Sea. The uncertainty of $C_{ant}$ concentrations is 5 $\mu$mol kg$^{-1}$, for DIC and for DIC$_{nat}$ it is 4 $\mu$mol kg$^{-1}$. Errors for the inventories were estimated by propagating the
uncertainties of the layer thicknesses and the concentrations through the water column.



## 5   Results

Over the 24-year period considered here, the frequency and the amplitude of the mean-winter NAO changed significantly (Hurrell and Deser, 2009). The convection in the SPG during winter, which is driven by the large-scale atmospheric circulation (Lazier et al., 2002; Pickart et al., 2003b; Yashayaev et al., 2007), has varied in strength and extent accordingly. Three

distinctive time periods can be characterized from 1991 to 2015 by different levels of convective activity in the Irminger Sea. In the first period, from 1991 to 1997, several consecutive positive NAO winters led to extensive deep convection in the entire SPG (Pickart et al., 2003a, b). In the second period, from 2000 to 2007, the NAO was in a more neutral state and only shallow convection occurred in the Irminger Sea. In the third period from 2008 to 2015, three deep convective events took place in the Irminger Sea, in 2008, 2012 and 2015 (Våge et al., 2008; de Jong et al., 2012; Fröb et al., 2016; de Jong and de Steur, 2016).

The DIC, $DIC_{nat}$ and $C_{ant}$ inventory changes are presented with respect to these three periods and for the entire 24 years of observations in the following sections.

The temporal change in DIC, $DIC_{nat}$ and $C_{ant}$ concentration in the Irminger Sea is clearly visible in Fig. 5, which shows interpolated cruise station data for 1991, 1997, 2007 and 2015, start and end years of the three periods considered here. The

increase in DIC is evident throughout the basin. From 1991-1997, cLSW with a low DIC-signature dominates the basin, but a tongue of older IW over the Reykjanes Ridge transports relatively high DIC concentrations into the Irminger Sea. By 2015 the DIC concentration had increased by at least 10 $\mu$mol kg$^{-1}$ compared to 1991, as visualized by the disappearance of the 2150 $\mu$mol kg$^{-1}$ contour line (Fig. 5a). Temporal changes in $DIC_{nat}$ concentration (Fig. 5b) are small compared to those in DIC and less systematic. The $C_{ant}$ concentration increases over time, not only at the surface, but over the entire water column, which is

indicated by the disappearance of the 20 $\mu$mol kg$^{-1}$ contour line below 1500 m from 1991 to 2015 (Fig. 5c).

The column inventory time series of DIC, $DIC_{nat}$ and $C_{ant}$, shown in Fig. 6, quantifies this large temporal change in the Irminger Sea sections. Typically, the DIC inventory increased by 1.43±0.17 mol m$^{-2}$yr$^{-1}$, from 1991 to 2015, from approximately 5645 to 5685 mol m$^{-2}$. The $C_{ant}$ storage rate was 1.84±0.16 mol m$^{-2}$yr$^{-1}$ in the same time period, this is

larger than the rate of DIC inventory change. At the same time, the $DIC_{nat}$ inventory decreased at a rate of -0.57±0.22 mol m$^{-2}$yr$^{-1}$. Therefore, the annual change in the DIC inventory is mainly driven by the large $C_{ant}$ storage rate, but partially offset by the loss in $DIC_{nat}$ inventory. The variability of the DIC and $C_{ant}$ inventories over the 24-year period is of similar magnitude, as indicated by the error of the slope in Fig. 6, whereas the $DIC_{nat}$ inventory varies slightly more. It is notable that the $C_{ant}$ inventory increased sharply from 2012 to 2015, while there was a comparably large decline in the $DIC_{nat}$ inventory.

This is not an artefact of the method, but can be explained by the fact that the 2015 data were obtained during active convection in the Irminger Sea (Fröb et al., 2016). In contrast to that, the peak in 2005 in the $C_{ant}$ inventory cannot be explained by a similar mechanism, therefore it is unlikely a real signal but rather might reflect the true error or reveal measurement bias.



## 5.1 SWT distribution

The layer thickness of the Irminger Sea SWTs from 1991 to 2015 is presented in Fig. 7. Because their individual contributions are small, the upper ocean SWTs, i.e. NACW, ISW, and SPMW, are combined and titled upper waters (UW). Based on the composite analysis, the contributions of uNEADW and IcSW are divided up and added to cLSW, IW and ISOW. MW and lNEADW only appear in the Iceland Basin and are not included for further analysis. Therefore, not all 11 SWTs used for the eOMP analysis are shown, but only UW, IW, uLSW, cLSW, DSOW and ISOW. Since the FOUREX section, occupied in 1997, was located further south than the AR07E section, covered that year by 06MT19970707, the SWT distribution differs slightly between the two cruises. At the FOUREX section, the ISOW layer is on average 50 m and the SAIW layer about 15 m thicker than further north, while the ISW layer is about 47 m and the SPMW layer 19 m thicker for 06MT19970707 than at the FOUREX section. For the other SWTs, differences are smaller than 8 m. Relatively speaking, 50 m is less than 2 % of the entire water column, so that the discrepancy between the two cruises is small compared to the mean depth of the Irminger Sea.

The distribution of SWTs in the Irminger Sea, as shown in Fig. 7, changes substantially from 1991 to 2015. The overall trend is indicated, but the rates of change are often larger, if sub-periods are considered. For example, the IW layer thickens slightly from 1991 to 2015 at a rate of $6.5\pm2.0$ m yr$^{-1}$, but there is little change before 2004, while after, the layer thickness increase is much stronger. The uLSW layer thickness increases by an average of $24.1\pm8.7$ m yr$^{-1}$, but it is very thin before 1997 and the major build up occurs after 2004. In particular, after deep convection in the Irminger Sea in 2008, 2012 and 2015, the layer thickness of uLSW increases substantially. The cLSW layer shows the largest changes in thickness at a loss rate of $-54.0\pm4.3$ m yr$^{-1}$ from 1991 to 2015. The convective activity in the SPG in 1991 to 1997 lead to extensive production of cLSW. After that, the cLSW layer was not renewed and strongly diminished until 2015. In contrast to that, the DSOW layer decreases only little in thickness over the 24-year period covered by the data. The ISOW layer thins by $-9.9\pm3.1$ m yr$^{-1}$. Over the entire period from 1991 to 2015 the UW layer thickness increases at a rate of $26.2\pm4.8$ m yr$^{-1}$, but it has essentially a constant thickness from 1991 to 2000, then thickens until 2007 and decreases in thickness after that. Overall, the change of the distribution of the main SWTs is well captured by the eOMP analysis. Especially the transformation of the two LSW classes seems to match the observations well (Yashayaev et al., 2007).

The mean layer thickness for each SWT in the time periods considered here are summarized in Table 3. The main variability in the distribution of water masses is created by layer thickness changes of cLSW, uLSW and UW. With only small changes over the 24-year period, DSOW, ISOW and IW occupy a little more than a third of the Irminger Sea sections. In the mid-1990s, the cLSW layer occupied close to 50 % of the entire water column, which left thin uLSW and UW layers, corresponding to less than 7 % and around 8 %, respectively, of the entire water column. As cLSW was advected out of the Irminger Sea, while not being re-formed by convection, that layer was replaced mainly by UW in the early 2000s. In 2004, cLSW occupied 37 % of the water column and UW 24 %, while the uLSW layer only accounted for 4 %. With the recurring convection events between 2008 and 2015, uLSW was formed more frequently while displacing UW as well as the remainder of the cLSW layer.



The measurements from winter 2015 reveal that by then the fraction of cLSW was as low as 17 %, and that of UW only 5 %, but that uLSW occupied 45 % of the entire water column.

## 5.2 Archetypal concentration changes

Within the SWTs, the archetypal concentrations of carbon species change over time due to e.g. remineralization of organic mat-

ter, which add $DIC_{nat}$, or air-sea gas exchange, which increases $C_{ant}$ in water masses that are in contact with the atmosphere. The archetypal concentrations of DIC, $DIC_{nat}$ and $C_{ant}$ from 1991 to 2015 are shown in Fig. 8, for the same SWTs as in Fig. 7, and are also summarized in Table 3. In all SWTs, DIC is similar, except for UW, which have a more variable DIC that is generally lower compared to all other SWTs. The older SWTs in the deep ocean have higher $DIC_{nat}$ concentrations and lower $C_{ant}$ concentrations compared to SWTs that have been ventilated more recently. Therefore, the deep SWTs, namely ISOW and

DSOW, have the highest archetypal concentrations of $DIC_{nat}$ (both 2136 $\mu$mol kg$^{-1}$) and the lowest archetypal concentrations of $C_{ant}$ (22 $\mu$mol kg$^{-1}$ and 20 $\mu$mol kg$^{-1}$), respectively. In contrast, UW have the lowest archetypal concentrations of $DIC_{nat}$ (2112 $\mu$mol kg$^{-1}$) and the highest archetypal concentrations of $C_{ant}$ (40 $\mu$mol kg$^{-1}$), which are almost twice as high as in the deep SWTs.

For all SWTs, the rate of change in the archetypal DIC concentration is similar, except for ISOW, where it is slightly smaller. The archetypal concentration change over time for $DIC_{nat}$ is not statistically different from zero for most SWTs, apart from cLSW, where the $DIC_{nat}$ concentration increases by 0.30±0.06 $\mu$mol kg$^{-1}$yr$^{-1}$. In contrast, the archetypal $C_{ant}$ concentration increases significantly in all SWTs over time. Further, the increase rate of the archetypal DIC concentration is not statistically different from the rate of increase of the archetypal $C_{ant}$ concentration. This indicates that the increase in DIC can be explained

by the input of $C_{ant}$ to the entire water column. This is true for all SWTs, except cLSW. For this water mass the increase in the archetypal $C_{ant}$ concentration contributes by 0.31±0.07 $\mu$mol kg$^{-1}$yr$^{-1}$ to the increase in DIC, which is only half of the DIC concentration increase. This is because $DIC_{nat}$ accumulates as cLSW ages, while at the same time, a smaller fraction of $C_{ant}$ is added to this water mass due to less frequent ventilation.

## 5.3 DIC storage rate decomposition

The decomposition of the inventory rates of change reveals the contribution of changes in the SWT distribution and of changes in the concentration within these SWTs to the trends in DIC and its natural and anthropogenic components. Figure 9 summarizes the storage rates of DIC, $DIC_{nat}$ and $C_{ant}$ over the entire 24-year time period and for the periods 1991-1997, 2000-2007 and 2008-2015. The first bar shows the total storage rate summed over all SWTs. The second bar shows the concentration driven storage rate and the third bar the layer thickness-driven storage rate. In theory, the first bar should be the sum of the last two

bars. However, since the storage rates have been calculated using a linear regression over only a small number of data points, the residuals can become quite large, this is especially the case for the shorter time periods. Nevertheless, some conclusions can be drawn. The increase of the DIC inventory from 1991 to 2015 is driven by the increase in the $C_{ant}$ inventory, partially offset by a decrease in $DIC_{nat}$ inventory. The rise in the $C_{ant}$ inventory is primarily due to a rise in $C_{ant}$ concentration, but the





contribution from layer thickness-driven changes is also significantly positive (Fig. 9a). While the rise in $C_{ant}$ concentration occurs in all SWTs (Fig. 8), the contribution from the latter factor appears mostly driven by the increase in the thickness of uLSW (Fig. 7), which is rich in $C_{ant}$ (Fig. 8). The decrease in the $DIC_{nat}$ inventory is the result of the layer thickness-driven reduction, which is larger than the concentration driven increase (Fig. 9a). This can be attributed to the replacement of cLSW

with uLSW and UW, which both have lower concentrations of $DIC_{nat}$.

The variations at subdecadal timescales can be understood in terms of the convective activity in the Irminger Sea, although with larger uncertainty. Figure 9b shows the decomposed trends from 1991-1997, a period when convective activity was high in the SPG. The total DIC storage rate is driven by the $C_{ant}$ storage rate, while the changes in the $DIC_{nat}$ inventory are not

significantly different from zero. The $C_{ant}$ storage rate is mainly driven by increasing concentrations, which occur in all SWTs (Fig. 8).

From 2000-2007, although the $C_{ant}$ storage rate of 2.14±0.49 mol m$^{-2}$yr$^{-1}$ was similar to that of the preceding period, the DIC storage rate was much smaller because of the large loss of $DIC_{nat}$. 2000-2007 has smallest DIC storage rate of all periods

considered, 0.65±0.39 mol m$^{-2}$yr$^{-1}$, compared to 2.53±1.24 and 1.93±0.20 mol m$^{-2}$yr$^{-1}$ for the earlier and later periods, respectively. Also, unlike the other periods, changes in layer thickness and concentrations were almost equally important for the $C_{ant}$ storage rate. The $DIC_{nat}$ storage rate, on the other hand, was almost entirely driven by changes in layer thickness: -1.20±0.29 mol m$^{-2}$yr$^{-1}$ of a total of -1.63±0.98 mol m$^{-2}$yr$^{-1}$. From Fig. 7 and Fig. 8 these features appear to be the result of uLSW and UW replacing cLSW. Their larger concentrations of $C_{ant}$ leads to the relatively large layer thickness-driven

storage increase, while the advection of $DIC_{nat}$-rich cLSW out of the Irminger Sea leads to the negative layer thickness-driven decrease of the $DIC_{nat}$ inventory. The negative concentration-driven storage rate appears primarily driven by the loss of $DIC_{nat}$ from uLSW and UW, outweighing the increase of $DIC_{nat}$ in the cLSW.

For the last period from 2008-2015 (Fig. 9d), the deep convection in 2015 had a large impact on all storage rates, as well

as their uncertainty estimates, i.e. the exceptional changes in 2015 incurs a large uncertainty on the regression slopes for this time period. The DIC storage rate is 1.93±0.20 mol m$^{-2}$yr$^{-1}$ and the result of a large $C_{ant}$ storage rate, the largest in all three periods considered, offset by a negative $DIC_{nat}$ storage rate. Both of these are primarily the result of changes in concentrations, indicative of the ventilation that occurred. As evaluated from Fig. 8, the $C_{ant}$ increase is greatest in IW, uLSW and UW, while the loss of $DIC_{nat}$ occurred primarily in the IW and uLSW. However, also cLSW appears to be affected by the most recent

event in 2015.

## 6   Discussion

The data that have been collected in the Irminger Sea over the past decades provide unequivocal evidence for climate forcing of the carbon cycle in this oceanic region. Over long time scales, the steady trend due to uptake of anthropogenic $CO_2$ clearly





dominates, but at shorter time scales, it varies and can also be significantly masked by variability of $DIC_{nat}$. In particular, this was the case in the period from 2000 to 2007, when the negative $DIC_{nat}$ storage rate partially offset the increasing $C_{ant}$ storage, resulting in a DIC storage rate that was only about a third of the storage rates in the preceding and later time periods considered here. In that period when convection was shallow, the replacement of relatively $DIC_{nat}$ rich cLSW with relatively

$DIC_{nat}$-poor UW and uLSW led to the loss of $DIC_{nat}$. Ageing might have increased the $DIC_{nat}$ concentration, but the water masses in which these processes occur were flushed out the study area. Climate feedback mechanisms that involve natural carbon cycling in the ocean are relevant to elucidate. While for example Zunino et al. (2015) found steady-state conditions for the natural carbon cycle in the entire eastern SPNA between 2002 and 2010, this was not the case for the Irminger Sea from 1991-2015. A possible explanation for this discrepancy could be the longer time period considered here, or that if both the

Irminger Sea and the Iceland Basin are regarded, inventory changes might cancel each other out. This was shown to be the case for $C_{ant}$ by Steinfeldt et al. (2009), as a consequence of opposite changes in cLSW volume east and west of the Reykjanes Ridge.

The results presented here do not indicate a consistent response in the storage rates for anthropogenic $CO_2$ to the NAO. The

storage rates were similar for the first two periods, with predominantly high and low values of the NAO index, while it was clearly larger for the last period, with predominantly high NAO index. In that time period, the $C_{ant}$ storage rate increased by a factor of 1.6 compared to the period from 2000 to 2007 (Fig. 9). The lack of change in $C_{ant}$ storage rates between the two first periods, contrasts with the results of Pérez et al. (2008), who found that the storage rates were low from 1997-2006. This is a result of the differences in data: when using the same cruises and the same periods of time (i.e 1997-2006 for the middle

period) as in Pérez et al. (2008) to estimate $C_{ant}$ storage rates, we estimated a significant decline in $C_{ant}$ storage rates for the middle period compared to the first ($C_{ant}$ storage rates: 2.78±0.28 mol m$^{-2}$yr$^{-1}$ for 1991-1997: 1.06 $pm$0.47 mol m$^{-2}$yr$^{-1}$ for 1997-2006).

The DIC storage, on the other hand, do show a consistent response to the NAO index, it is larger in the periods with pre-

dominantly high NAO index winters (the first and the last) than in the middle period, with a predominance of low NAO index winters. This response was also found when using the same cruises and time periods as Pérez et al. (2008) to calculate the storage rates. Altogether, this shows that while calculation of $C_{ant}$ storage rates over small time periods is sensitive to data selection, calculation of DIC storage rates is not. This is not unreasonable, as estimates of $C_{ant}$ inventories involves more variables (for example AOU and alkalinity), which increases the risk of introducing sampling or measurement biases. Regard-

less, convective events increase the storage of $C_{ant}$, this is clearly demonstrated for the 2015 event, also presented in detail by Fröb et al. (2016), who also used an independent approach for estimating $C_{ant}$. Most likely because the thick layer of $C_{ant}$ rich cLSW was mostly established in 1991 when the first data were collected, no large storage rate is observed for the early NAO positive period included here (Yashayaev et al., 2007).



Tanhua and Keeling (2012) calculated North Atlantic column inventory changes of DIC using data extracted from GLO-DAPv1 (Key et al., 2004) and CARINA (Key et al., 2010). For obtaining further insight on the governing processes, they also included $O_2$, AOU and $DIC_{abio}$, which is DIC corrected for the fraction of remineralised carbon and is closely related to anthropogenic carbon. However, their inventories were only estimated over the upper 2000 m of the water column. For DIC, the storage rate in the Labrador and Irminger Seas combined was estimated to be 0.57 mol $m^{-2}yr^{-1}$ (Tanhua and Keeling, 2012). This is only a third of our estimates of 1.43±0.17 mol $m^{-2}yr^{-1}$. This difference is likely the result of the different depth ranges, region and time periods evaluated. It is, however, noteworthy, that the $DIC_{abio}$ storage rate calculated by Tanhua and Keeling (2012) is not significantly different from zero. The entire increase in DIC inventory is explained by an increase of the remineralised fraction, as determined from AOU and their assumed C:O ratio. This implies either that there is no storage of $C_{ant}$ in the region or that any increase in $C_{ant}$ is completely offset by reduced $CO_2$ solubility.

In order to compare to the Tanhua and Keeling (2012) results here, the $DIC_{abio}$ storage rates as well as $O_2$ and AOU storage rates are shown in Fig. 10 for the same periods as in Fig. 9. For the Irminger Sea cruise data from 1991-2015, the $DIC_{abio}$ storage rate is 1.40±0.17 mol $m^{-2}yr^{-1}$, and explains the DIC storage rate almost entirely. The negligible trend in AOU explains the lack of difference between the DIC and $DIC_{abio}$ storage rates. The estimated $DIC_{abio}$ storage rate is lower than the $C_{ant}$ storage rate (1.92±0.17 mol $m^{-2}yr^{-1}$, Fig. 6), probably due to a decline in preformed DIC values, i.e. a loss of solubility as a consequence of the positive surface temperature trends in the Irminger Sea from 1991-2015 (Stendardo and Gruber, 2012; Maze et al., 2012). The long-term warming trends in the surface ocean and thus the decreasing $O_2$ solubility leads to a deoxygenation of -0.8±0.3 mol $m^{-2}yr^{-1}$ over the 24-year period, which is consistent with the suggested loss in preformed DIC.

On shorter timescales (Fig. 10b-c) all variables show large variations in the storage rates consistent with the already discussed changes in hydrography. In contrast to the findings of Tanhua and Keeling (2012), DIC and $DIC_{abio}$ storage rates are positive over all time periods. This might be partly due to the increased data coverage in our study. Over the entire time period, $DIC_{abio}$ underestimates $C_{ant}$ by about 25%. However, if the time period is too short, $DIC_{abio}$ becomes much more variable, because it includes solubility effects. For example, from 2000-2007, the $DIC_{abio}$ underestimates the $C_{ant}$ storage rate by 40 %, (compare Fig. 9c and 10c), while from 2008-2015, the $DIC_{abio}$ overestimates the $C_{ant}$ storage rate by 20 % (compare Fig. 9d and 10d).

Similar to the $C_{ant}$ storage rates, both periods before and after 1997 show a loss in $O_2$, despite stronger convection in the early 1990s. Again, this is attributed to the fact that cLSW was mostly formed before 1991 (Yashayaev et al., 2007). The mean $O_2$ saturation degree over the entire water column was 89 % in the period from 1991 to 1997, which resulted in a high $O_2$ inventory of 750±4 mol $m^{-2}$, indicative of the recent ventilation. After 1997, no re-ventilation took place and the mean $O_2$ saturation dropped 2 %, while the inventory decreased to 732±4 mol $m^{-2}$. From 2008 to 2015, the $O_2$ saturation values increased to 89 % again due to the strong inventory increase of 4.3±2.6 mol $m^{-2}yr^{-1}$, which was mainly driven by the deep



convection in 2015 (Fröb et al., 2016). In fact, the convection in 2015 was strong enough to restore the $O_2$ inventory levels so that the mean inventory from 2008 to 2015 was 749±4 mol m$^{-2}$, the level of the well-ventilated early 1990s.

## 7   Conclusions

The repeat observations within the Irminger Sea show significant changes in total, natural and anthropogenic $CO_2$ inventories

from 1991 to 2015 with large interannual variability in the natural component. The eOMP method results and the decomposition of the inventory changes give valuable insight into the driving mechanisms to interpret the observed variability. Overall, changes in layer thickness of the main water masses appear most important for the $DIC_{nat}$ inventory, while concentration change within these water masses is the key factor for $C_{ant}$. $C_{ant}$ is typically more important for changes in total DIC inventories than $DIC_{nat}$. While the DIC inventory changes show a clear signal associated to the NAO, for $C_{ant}$ the signal is less robust, especially before

and after 1997, likely because the data used here does not cover the period before 1991, when the thick layer of cLSW was formed. From 1991 to 2015 the mean $C_{ant}$ saturation over the entire water column increased from 52 % to 67 %, increasing the $C_{ant}$ inventory from 53±3 mol m$^{-2}$ to 117±3 mol m$^{-2}$, mainly driven by the most recent convection in 2015. Despite the negative trend in $O_2$ inventory from 1991 to 2015, the convection in 2015 was strong enough to replenish $O_2$ levels at depth, leading to a mean saturation of 89 % and an $O_2$ inventory of 749±4 mol m$^{-2}$, which was as high as in 1991. $C_{ant}$ is sensitive

to time period considered, while DIC appears more robust to sampling and measurement bias. Therefore, for a comprehensive view on carbon cycle feedback mechanisms, not only $C_{ant}$, but also natural and total DIC should be taken into account.

## Appendix A:  Salinity - Alkalinity relationship

The application of the surface relationship between salinity, temperature and $A_T$ by Lee et al. (2006) is tested for Irminger Sea cruise data in the entire water column. Further, the linear relationship between salinity and $A_T$ by Nondal et al. (2009) is

applied as well and compared to the Lee et al. (2006) relationship. For all cruise data between 1991 and 2015, the difference between measured and calculated $A_T$ is presented in Fig. A1. Both relationships perform reasonably well. For the Lee et al. (2006) relationship there is no bias with depth, but measured $A_T$ is slightly overestimated. The Nondal et al. (2009) relationship tends to overestimate the measured $A_T$ in the surface ocean and underestimate measured $A_T$ below 2000 m. Therefore the variance of the Nondal et al. (2009) relationship is much larger than for the Lee et al. (2006) relationship. Within this study,

the Lee et al. (2006) relationship was chosen in order to calculate $A_T$.

Further, the impact of the overestimated calculated $A_T$ on $C_{ant}$, $DIC_{nat}$ and DIC storage rates was tested. For that, 4.5 $\mu$mol kg$^{-1}$, which was the mean difference between measured and calculated $A_T$ based on the Lee et al. (2006) relationship, were added to the calculated $A_T$ for the 3 cruises, where only DIC was measured. The archetypal $C_{ant}$ concentrations for all SWTs

were less than 1 $\mu$mol kg$^{-1}$ smaller after the correction of $A_T$. No significant difference between the storage rates was evident.





## Appendix B: Location SWT

As a result of the eOMP, the fraction of SWTs in each water parcel is estimated. Figure B1 shows this fraction for the 1991 (a) and the 2015 (b) data for IW, uLSW, cLSW, DSOW, ISOW and UW, which is the sum over ISW, NACW and SPMW. Overall, the position of the water masses is well represented through time.

5  *Acknowledgements.* F. Fröb and A. Olsen appreciate funding from the SNACS project (229752), that is part of the KLIMAFORSK program of the Norwegian Research Council. E. Jeansson and S.K. Lauvset received funding from the NRC project VENTILATE (229791). F.F. Pérez and M.I. García-Ibáñez were supported by the Spanish Ministry of Economy and Competitiveness through the BOCATS (CTM2013-41048-P) project co-funded by the Fondo Europeo de Desarrollo Regional 2007-2012 (FEDER). This is a contribution to the BIGCHANGE project of the Bjerknes Center for Climate Research.



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

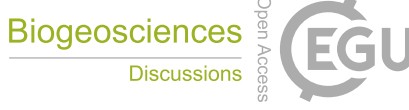

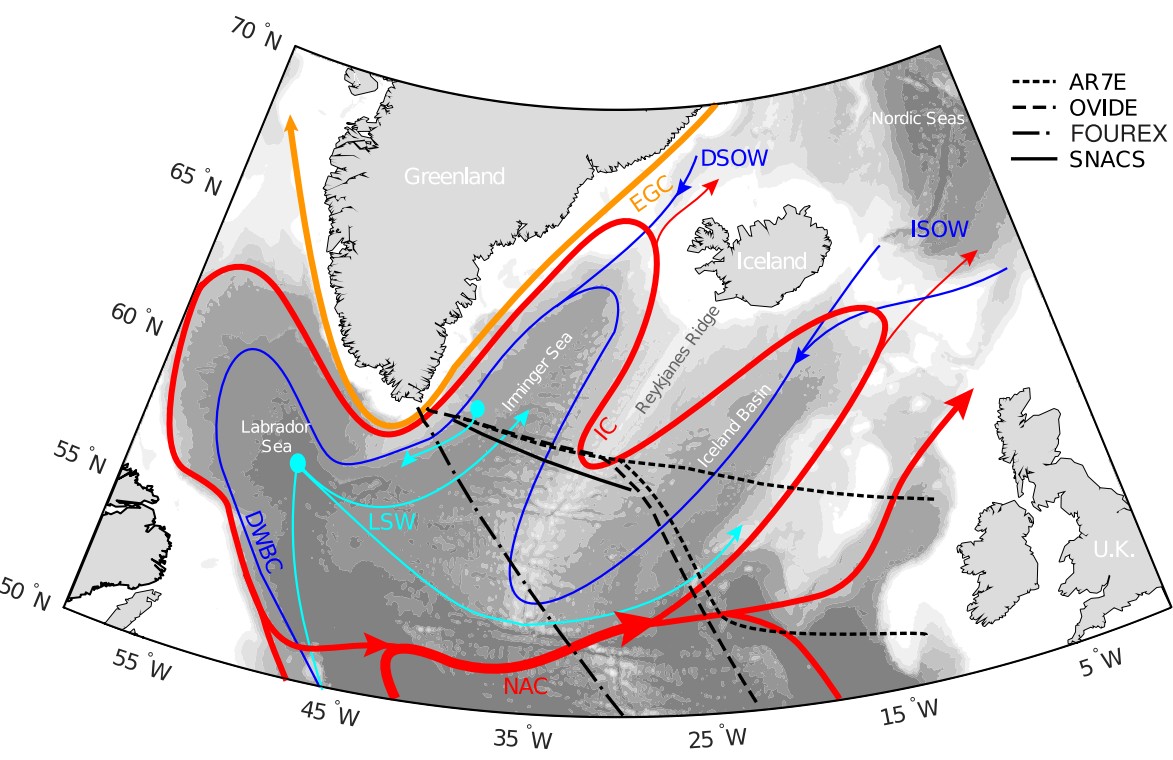

**Figure 1.** Schematic subpolar North Atlantic circulation. The location of the AR7E, OVIDE, FOUREX and SNACS lines are plotted in black on the bathymetry (500 m intervals). The branches of the North Atlantic Current (NAC) turning into the Irminger Current (IC) are shown in red and the East Greenland Current (EGC) is plotted in orange. The dark blue currents illustrate the spreading of the Iceland-Scotland Overflow Water (ISOW) and the Denmark Strait Overflow Water (DSOW) at depth, which jointly with the Labrador Sea Water (LSW), in cyan, contribute to the Deep Western Boundary Current (DWBC). Adapted from Lherminier et al. (2010) and Pérez et al. (2013).





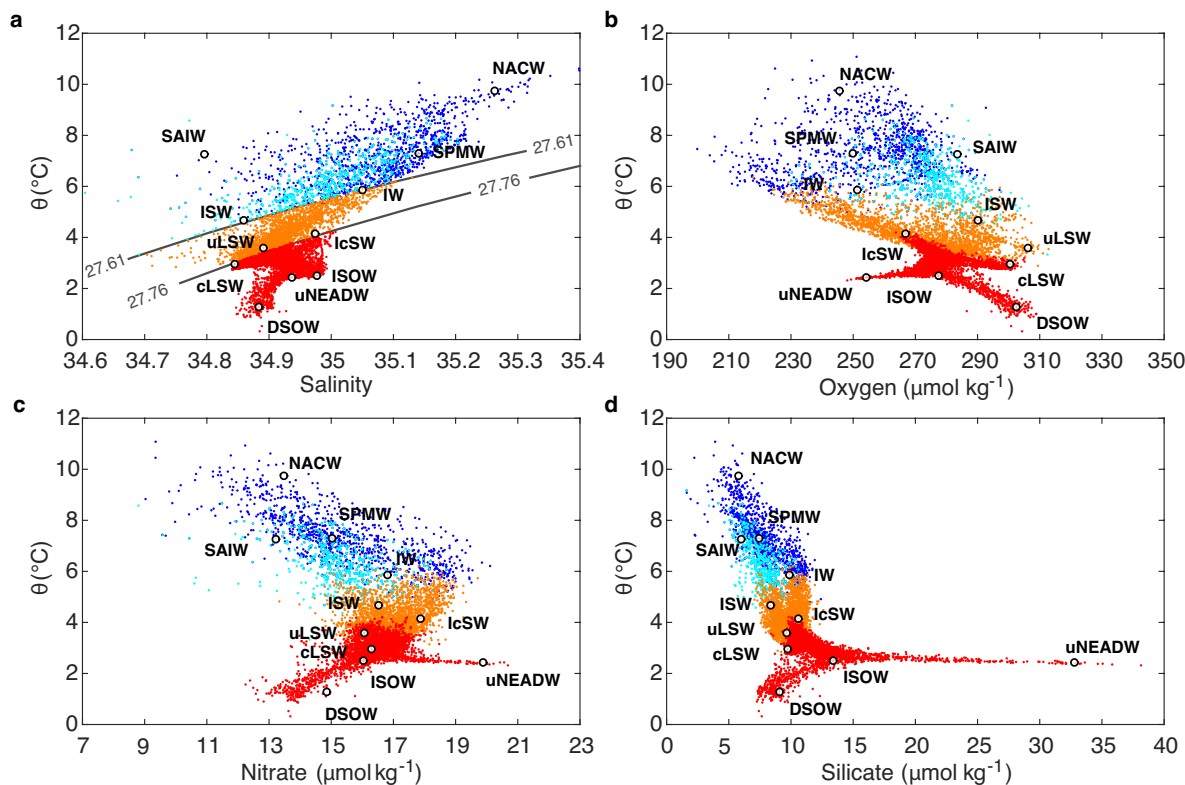

**Figure 2.** Source water type (SWT) parameters presented in a potential temperature - a) salinity, including potential density ($\sigma_0$) levels, b) oxygen, c) nitrate and d) silicate space for Irminger Sea cruise-data from 1991-2015. The colours represent different mixing figures for the eOMP analysis: red for the deep ocean ($\sigma_0 \geq 27.76$ kg m$^{-3}$), orange for the intermediate ocean ($27.61 \leq \sigma_0 < 27.76$ kg m$^{-3}$) and surface ocean ($\sigma_0 < 27.61$ kg m$^{-3}$), east of Reykjanes Ridge (dark blue) and west of Reykjanes Ridge region (cyan).





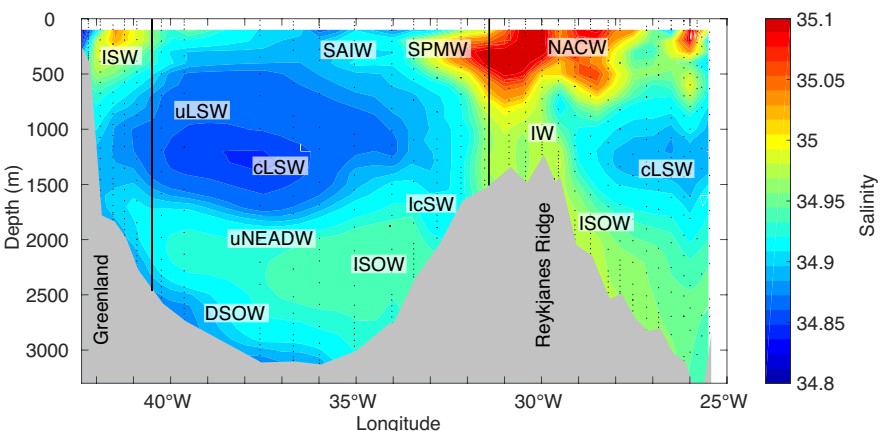

**Figure 3.** Vertical cross section through the Irminger Sea showing interpolated salinity based on the 1991 cruise data (06MT19910902). The approximate location of the main water masses of the eOMP analysis, shown in Appendix B, is illustrated: Subarctic Intermediate Water (SAIW), Intermediate Water (IW), classical and upper Labrador Sea Water (cLSW and uLSW), Denmark Strait Overflow Water (DSOW), upper Northeast Atlantic Deep Water (uNEADW), Iceland-Scotland Overflow Water (ISOW), Icelandic Slope Water (IcSW), Irminger Sea Water (ISW), Subpolar Mode Water (SPMW), North Atlantic Central Water (NACW). The longitudinal boundaries for the inventory estimates at 40.5° W and 31.5° W are shown (black lines).





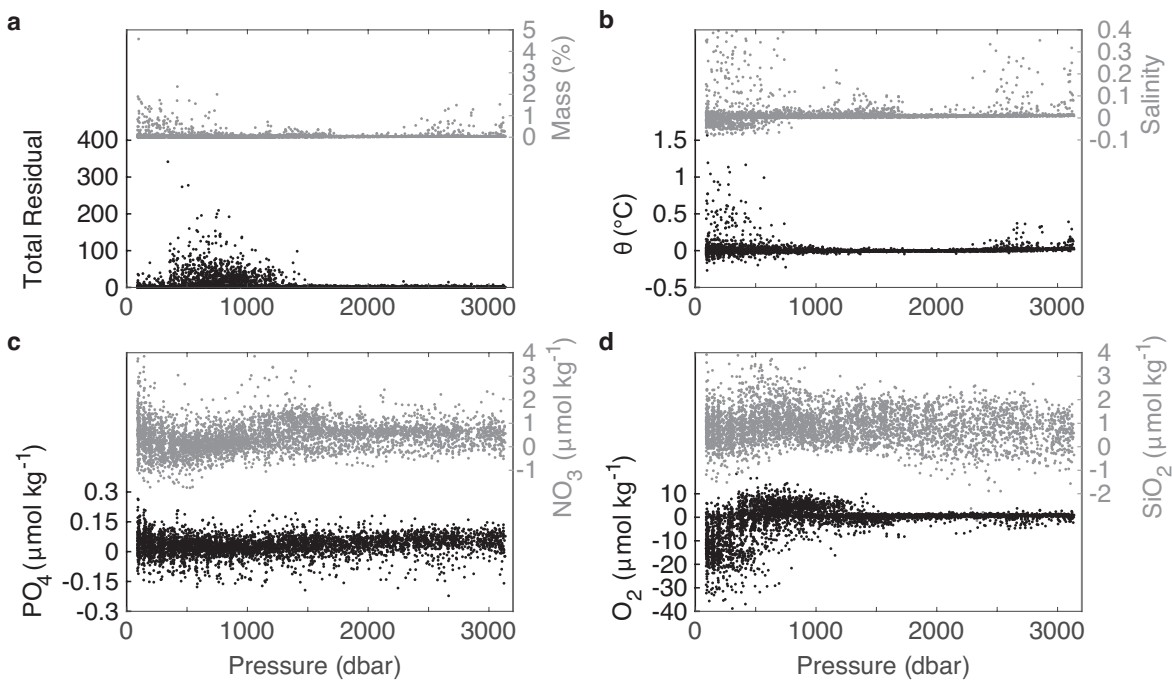

**Figure 4.** Residuals of the eOMP analysis for all Irminger Sea cruise-data from 1991-2015 for the a) total residual as the squared largest singular value for the set of residuals (García-Ibáñez et al., 2015), and the residual of mass conservation in %, b) residuals of potential temperature and salinity, c) residuals of phosphate and nitrate and d) residuals of oxygen and silicate with respect to pressure.





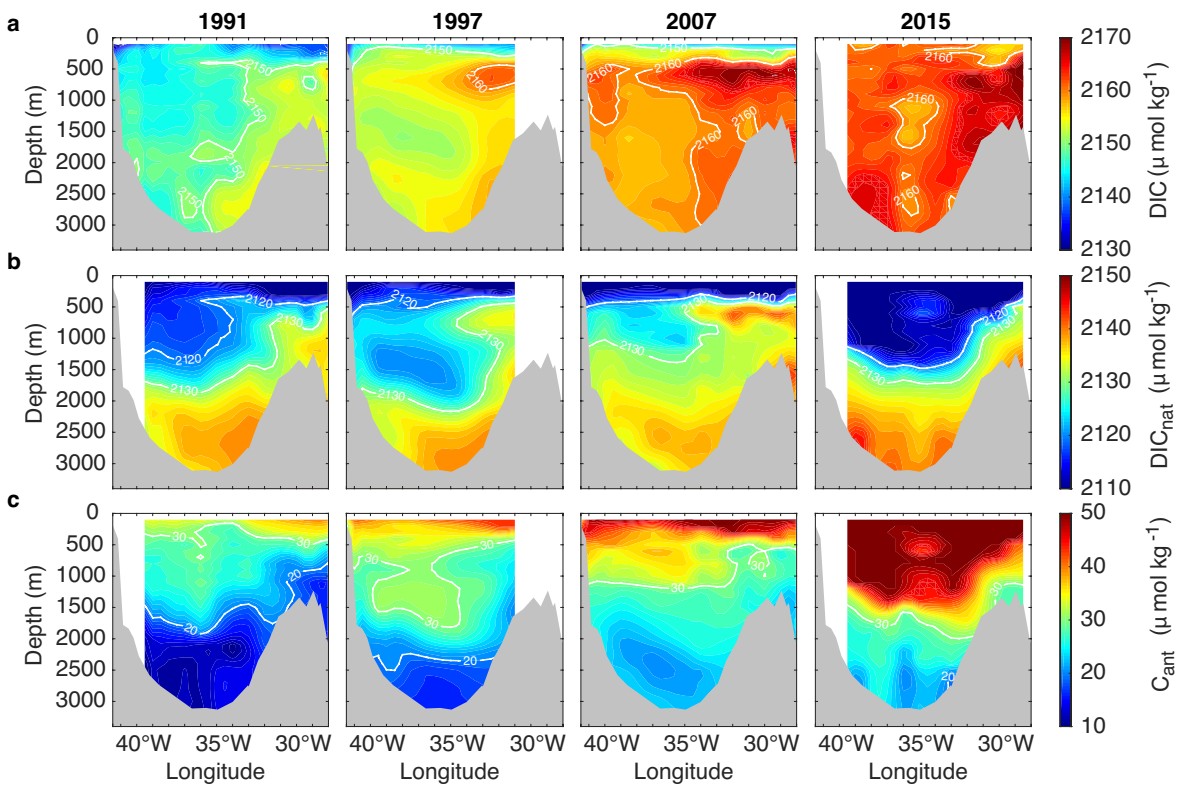

**Figure 5.** Vertical cross sections through the Irminger Sea showing interpolated cruise station data of a) DIC b) $DIC_{nat}$ and c) $C_{ant}$ concentration in 1991 (first row), 1997 (second row), 2007 (third row) and 2015 (last row). The white contour lines illustrate selected concentration levels. All panels have a the same span of values of 40 $\mu$mol kg$^{-1}$.





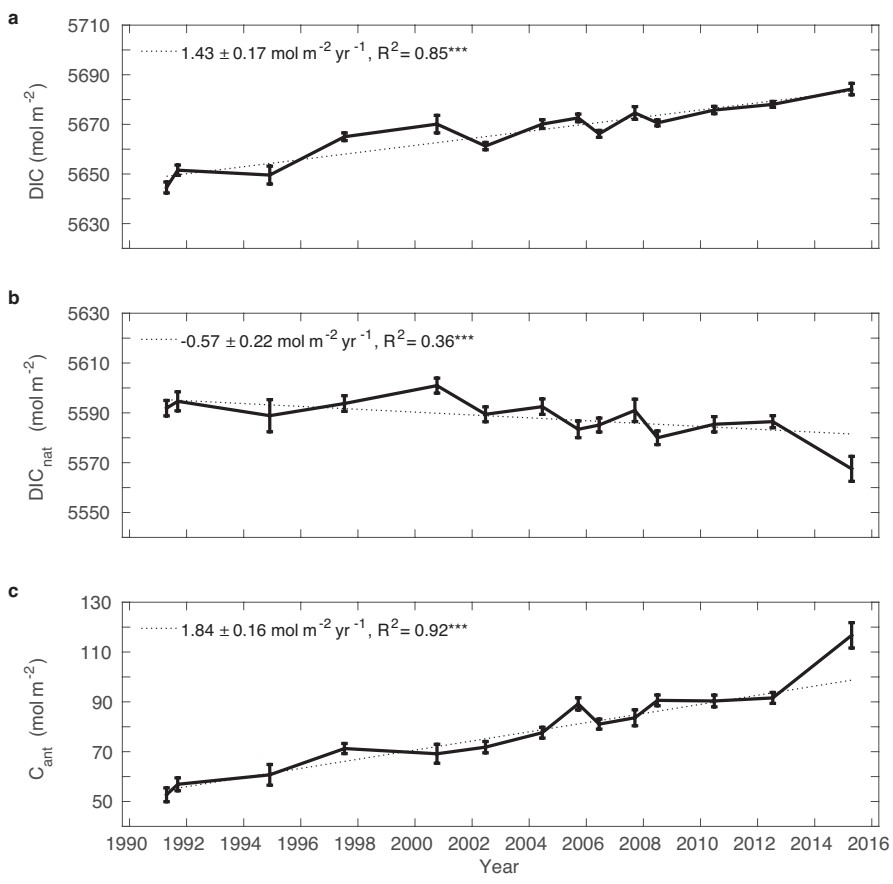

**Figure 6.** Total inventory for a) DIC, b) $DIC_{nat}$ and c) $C_{ant}$ for Irminger Sea cruise-data from 1991-2015. The rates of change between 1991 and 2015 are given, including the R-squared value of the linear regression model. The significance on the 99 % level (***) is indicated. Values of the two cruises in 1997 were averaged and are shown as one data point.





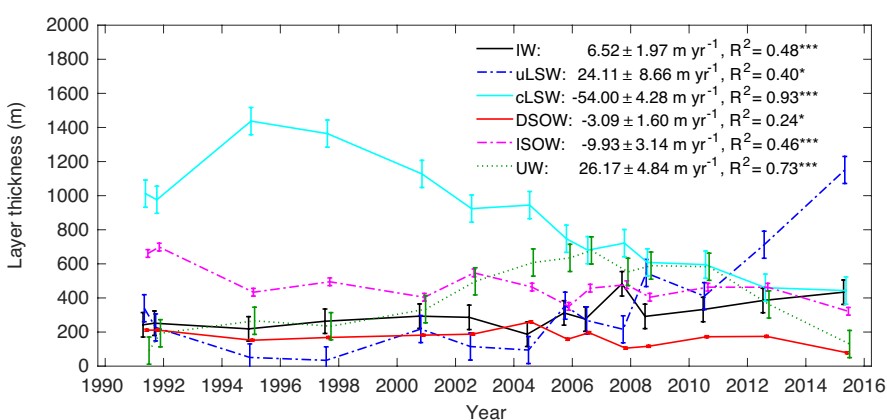

**Figure 7.** Layer thickness for IW, uLSW, cLSW, DSOW, ISOW and the sum over the upper ocean waters (UW) based on Irminger Sea cruise-data from 1991-2015. The mean depth between 40.5° W and 31.5° W is 2650 m. The error bars show the uncertainty based on a Monte Carlo simulation scaled to width and depth of the Irminger Sea. The rates of thickness change between 1991 and 2015 are given for all SWTs, including the R-squared value of the linear regression model. The significance on the 90 % level (*) or the 99 % level (***) is indicated. Values of the two cruises in 1997 were averaged and are shown as one data point. All markers are slightly offset in time for clarity.





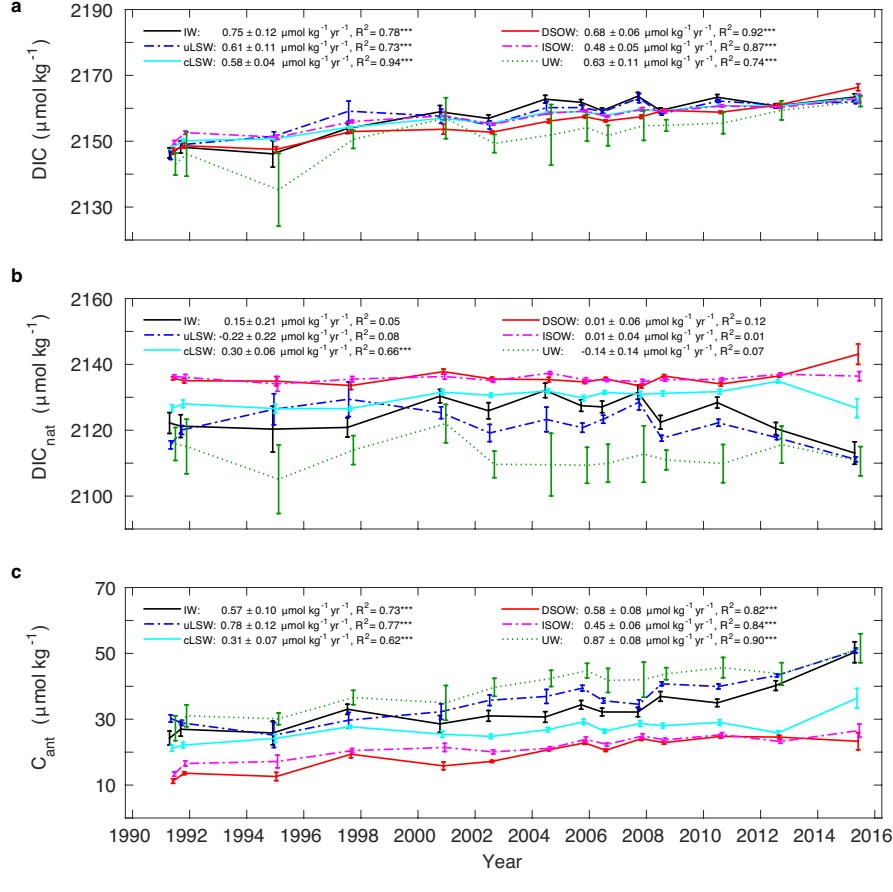

**Figure 8.** Archetypal concentration for IW, uLSW, cLSW, DSOW, ISOW and the sum over the upper ocean waters (UW) in the Irminger Sea from 1991 to 2015 for a) DIC, b) DIC$_{nat}$ and c) C$_{ant}$. The error bars represent $\sigma$. The storage rates between 1991 and 2015 are given for all SWTs, including the R-squared value of the linear regression model. The significance on the 90 % level (*) or the 99 % level (***) is indicated. Values of the two cruises in 1997 were averaged and are shown as one data point. All markers are slightly offset in time for clarity.




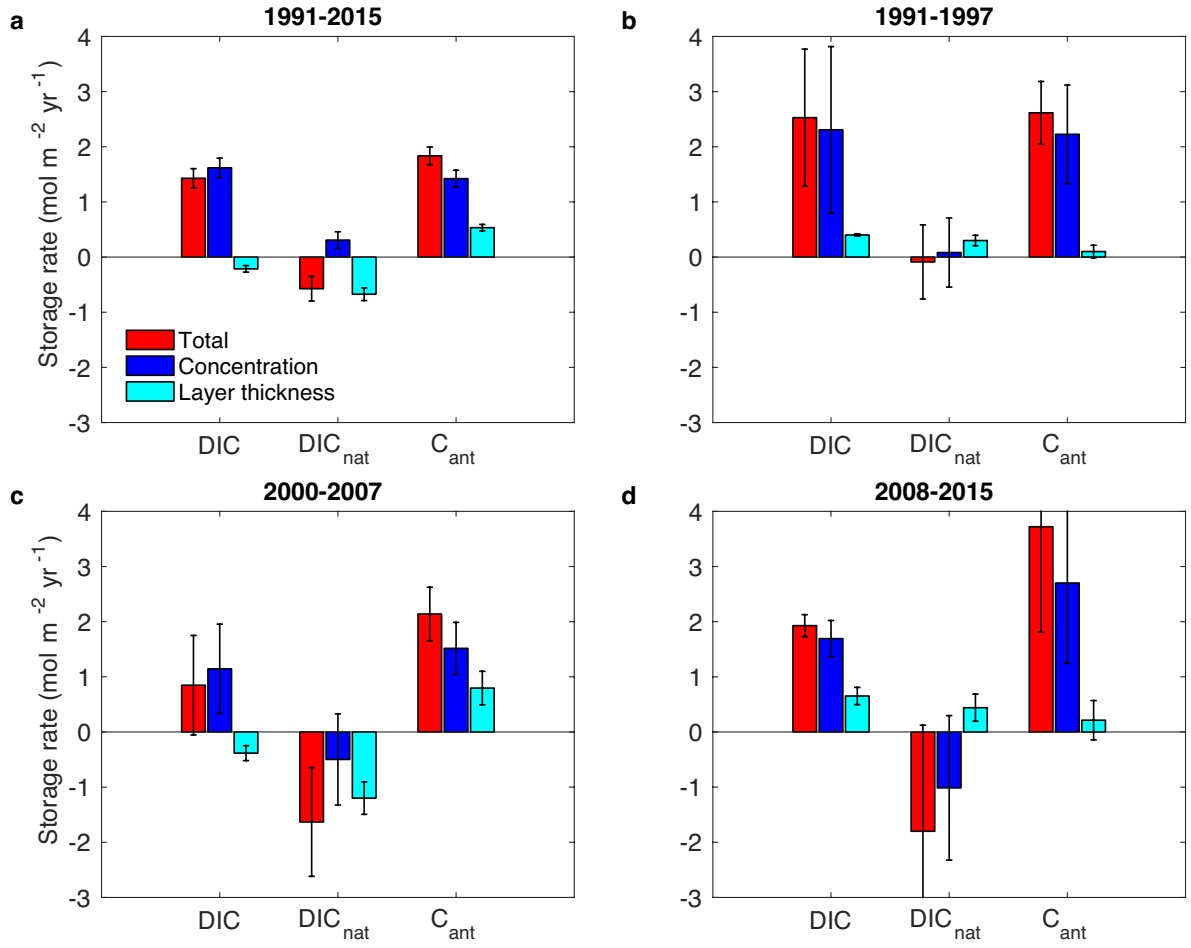

**Figure 9.** Decomposition of the total storage rates (red bars) into the concentration-driven storage rate (blue bars) and the layer thickness-driven storage rate rate (cyan bars) for DIC, $C_{ant}$ and $DIC_{nat}$ from a) 1991-2015, b) 1991-1997, c) 2000-2007 and d) 2008-2015. The error bars represent the error of the linear regression model.





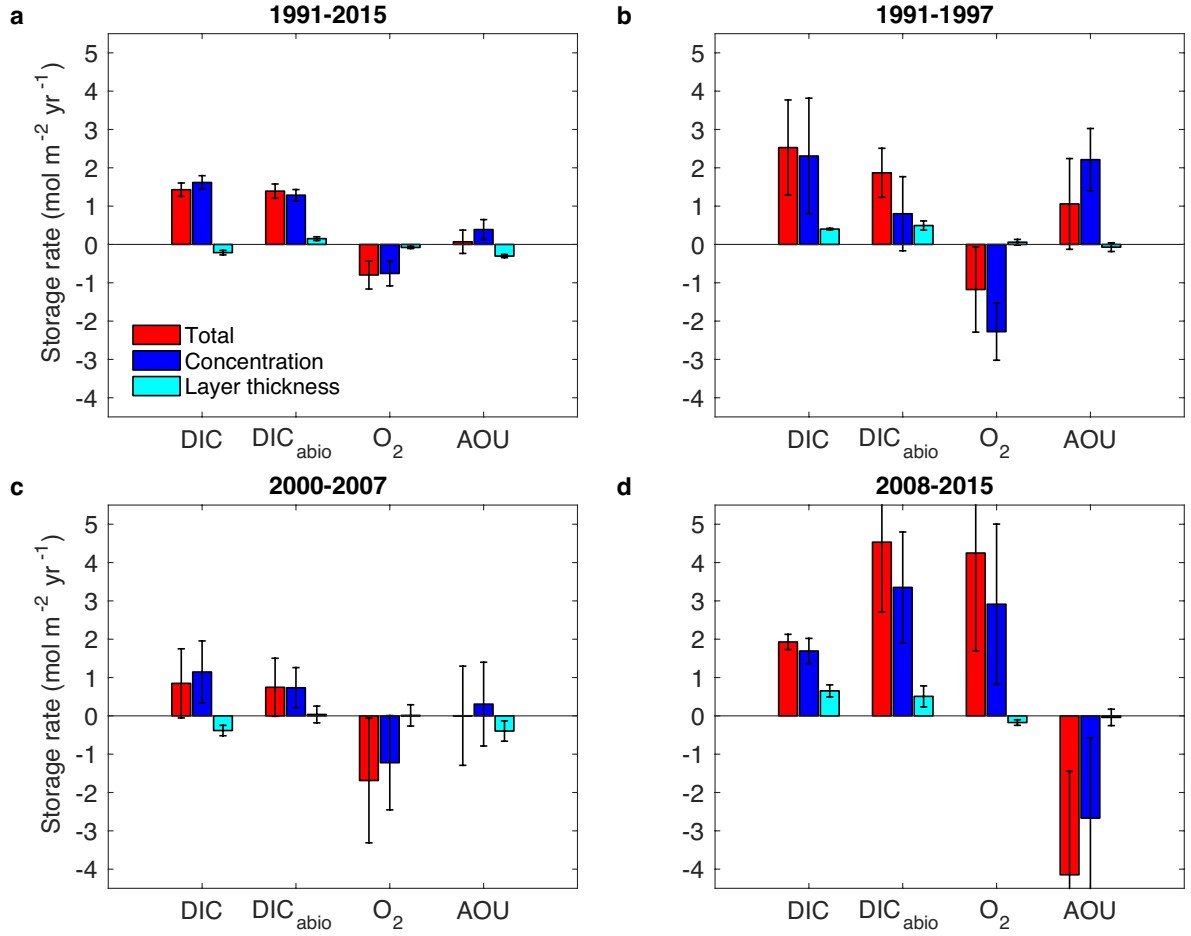

**Figure 10.** Decomposition of the total storage rates (red bars) into the concentration-driven storage rate (blue bars) and the layer thickness-driven storage rate (cyan bars) for DIC, $DIC_{abio}$, $O_2$ and AOU from a) 1991-2015, b) 1991-1997, c) 2000-2007 and d) 2008-2015. The error bars represent the error of the linear regression model.





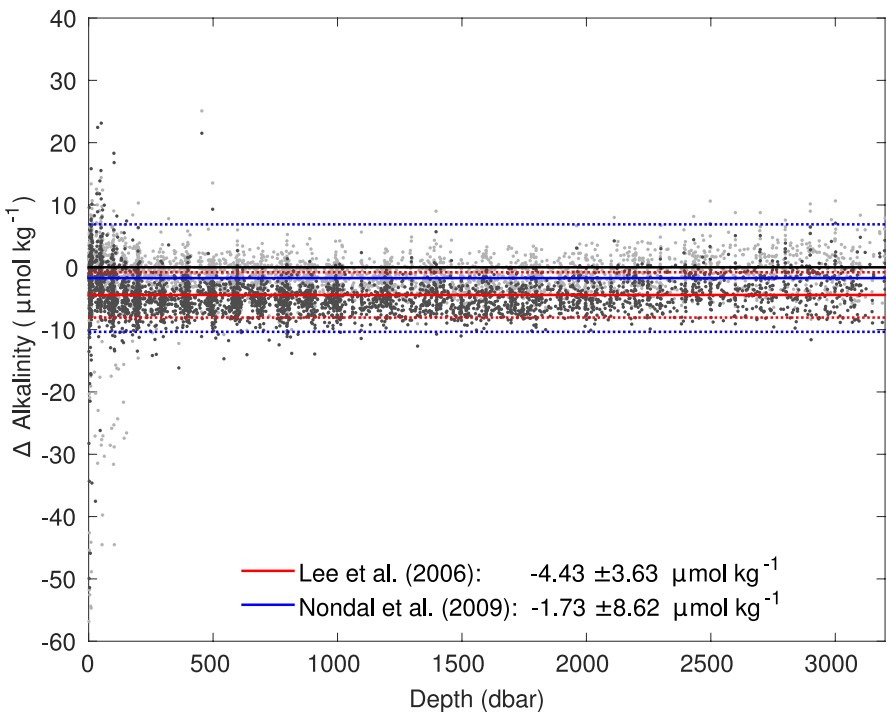

**Figure A 1.** Difference between measured and calculated alkalinity for Irminger Sea cruise data. The dark grey dots use the salinity-alkalinity relationship following Lee et al. (2006) and the light grey dots use the relationship following Nondal et al. (2009). The solid lines represent the mean difference and dotted lines ± one standard deviation.




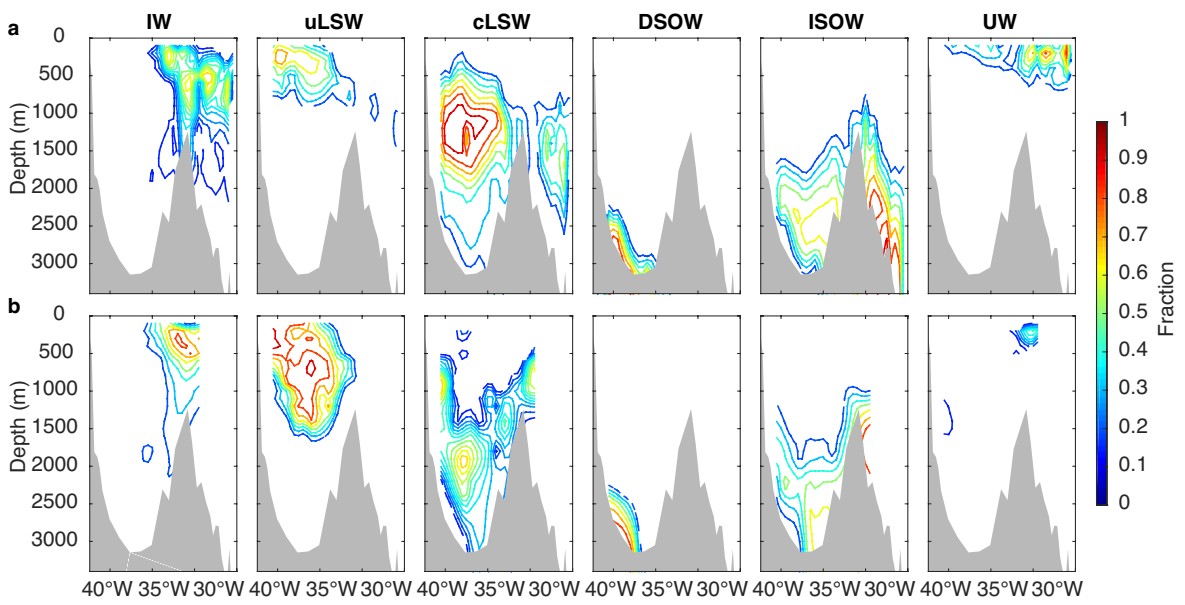

**Figure B 1.** Vertical cross section of fraction of water masses (IW, cLSW, uLSW, DSOW, ISOW and UW) in a) 1991 and b) 2015. Ratios below 0.1 are set to zero for clarity.





**Table 1.** Irminger Sea cruise information. The measured variables of the seawater $CO_2$ chemistry are indicated.

| Alias | Expocode | Month/Year | Ship | Data | Reference |
|---|---|---|---|---|---|
| AR07E | 64TR19910408 | 04-05/1991 | Tyro | DIC | Stoll et al. (1996) |
| A01E | 06MT19910902 | 09/1991 | Meteor | DIC, $A_T$ | Meincke and Becker (1993) |
| A01 | 06MT19941115 | 11-12/1994 | Meteor | DIC | Thomas and Ittekkot (2001) |
| FOUREX | 316N19970530 | 05-07/1997 | Knorr | DIC, $A_T$, pH | Johnson et al. (2003) |
| AR07W | 06MT19970707 | 07-08/1997 | Meteor | DIC, $A_T$ | Körtzinger et al. (1999) |
| AR7E | 64PE20000926 | 09-10/2000 | Pelagia | DIC | Yashayaev et al. (2007) |
| OVIDE 2002 | 35TH20020611 | 06-07/2002 | Thalassa | $A_T$, pH | Lherminier et al. (2010) |
| OVIDE 2004 | 35TH20040604 | 06-07/2004 | Thalassa | $A_T$, pH | Lherminier et al. (2010) |
| AR07E | 64PE20050907 | 09-10/2005 | Pelagia | DIC, $A_T$ | van Heuven et al. (2012) |
| OVIDE 2006 | 06MM20060523 | 05-06/2006 | Maria S. Merian | $A_T$, pH | Pérez et al. (2008) |
| AR07E | 64PE20070830 | 09/2007 | Pelagia | DIC, $A_T$ | van Heuven et al. (2013) |
| OVIDE 2008 | 35TH20080610 | 06-07/2008 | Thalassa | $A_T$, pH | Mercier et al. (2015) |
| OVIDE 2010 | 35TH20100610 | 06/2010 | Thalassa | $A_T$, pH | Mercier et al. (2015) |
| OVIDE 2012 | 29AH20120623 | 06-07/2012 | Sarmiento de Gamboa | $A_T$, pH | García-Ibáñez et al. (2016) |
| SNACS | 58GS20150410 | 04/2015 | G.O. Sars | DIC, $A_T$ | Fröb et al. (2016) |



**Table 2.** Source water type (SWT) parameters and their standard deviation used for the eOMP analysis for Subarctic Intermediate Water (SAIW), Intermediate Water (IW), classical and upper Labrador Sea Water (cLSW ans uLSW), Denmark Strait Overflow Water (DSOW), upper Northeast Atlantic Deep Water (uNEADW), Iceland-Scotland Overflow Water (ISOW), Icelandic Slope Water (IcSW), Irminger Sea Water (ISW), Subpolar Mode Water (SPMW) and North Atlantic Central Water (NACW). Definitions for Mediterraneum Water (MW) and lower Northeast Atlantic Deep Water (lNEADW) (*) from García-Ibáñez et al. (2015) are used only for the composite analysis but not in the eOMP. The weights for the equations are given. The mass weight is 150. The mean parameter residual is given (error). The last column gives an uncertainty estimate (per one) for the SWT contribution based on a Monte Carlo simulation.

| SWT | $\theta$ | S | $O_2$ | $PO_4$ | $NO_3$ | $SiO_2$ | PV | Uncertainty |
|---|---|---|---|---|---|---|---|---|
| | °C | | $\mu$mol kg$^{-1}$ | $\mu$mol kg$^{-1}$ | $\mu$mol kg$^{-1}$ | $\mu$mol kg$^{-1}$ | $10^{-8}$ m$^{-1}$s$^{-1}$ | |
| IW | 5.85± 0.25 | 35.05 ± 0.01 | 251.3± 1.7 | 1.17 ± 0.03 | 16.82 ± 0.56 | 9.87 ± 0.27 | 0.0547 ± 0.0184 | 0.030 |
| SAIW | 7.25± 0.07 | 34.80 ± 0.04 | 283.5± 6.1 | 0.86 ± 0.07 | 13.21 ± 1.53 | 6.01 ± 1.20 | 0.1100 ± 0.0158 | 0.005 |
| uLSW | 3.57± 0.12 | 34.89 ± 0.07 | 296.0± 9.3 | 1.06 ± 0.02 | 16.07 ± 0.35 | 9.68 ± 0.47 | 0.0009 ± 0.0169 | 0.034 |
| cLSW | 2.96± 0.01 | 34.85 ± 0.01 | 300.3± 1.0 | 1.07 ± 0.02 | 16.21 ± 0.23 | 9.74 ± 0.13 | 0.0003 ± 0.0186 | 0.034 |
| DSOW | 1.11± 0.16 | 34.88 ± 0.01 | 303.6± 1.0 | 0.95 ± 0.02 | 14.16 ± 0.35 | 9.23 ± 0.30 | 0.0479 ± 0.0184 | 0.003 |
| uNEADW | 2.42± 0.05 | 34.94 ± 0.01 | 254.3± 1.6 | 1.35 ± 0.04 | 19.88 ± 0.34 | 32.82 ± 2.46 | 0.0364 ± 0.0158 | $3.6*10^{-6}$ |
| ISOW | 2.50± 0.05 | 34.98 ± 0.01 | 277.5± 0.7 | 1.11 ± 0.03 | 16.04 ± 0.14 | 13.38 ± 0.51 | 0.0264 ± 0.0173 | 0.010 |
| IcSW | 4.13± 0.12 | 34.97 ± 0.01 | 267.0± 3.4 | 1.12 ± 0.01 | 17.87 ± 0.20 | 10.59 ± 0.30 | 0.0309 ± 0.0184 | 0.038 |
| ISW | 4.68± 0.30 | 34.86 ± 0.02 | 290.2± 1.0 | 1.07 ± 0.02 | 16.51 ± 0.90 | 8.35 ± 0.85 | 0.0364 ± 0.0158 | 0.017 |
| SPMW | 7.28± 0.25 | 35.14 ± 0.02 | 250.0± 0.2 | 0.98 ± 0.06 | 15.03 ± 1.00 | 7.44 ± 0.11 | 0.0479 ± 0.0096 | 0.016 |
| NACW | 9.74± 0.13 | 35.26 ± 0.02 | 245.7± 5.6 | 0.89 ± 0.01 | 13.47 ± 0.17 | 5.76 ± 0.42 | 0.0547 ± 0.0158 | 0.006 |
| MW* | 11.70 | 36.50 | 210 | 0.7 | 10.9 | 4.88 | - | - |
| lNEADW* | 1.98 | 34.90 | 252 | 1.5 | 22.6 | 48 | - | - |
| Weight | 25 | 15 | 8 | 2 | 2 | 1 | 3 | |
| Error | 0.006 | 0.015 | 0.881 | 0.031 | 0.421 | 0.927 | 0.001 | |





**Table 3.** Mean layer thickness and concentration of DIC, $DIC_{nat}$ and $C_{ant}$ in IW, uLSW, cLSW, DSOW, ISOW and UW in the time periods from 1991-2015,1991-1997, 2000-2007 and 2008-2015.

| Period | Variable | IW | uLSW | cLSW | DSOW | ISOW | UW |
|---|---|---|---|---|---|---|---|
| 1991-2015 | Thickness (m) | 301 | 338 | 860 | 170 | 474 | 411 |
| | DIC ($\mu$mol kg$^{-1}$) | 2158 | 2157 | 2157 | 2157 | 2155 | 2152 |
| | DIC$_{nat}$ ($\mu$mol kg$^{-1}$) | 2125 | 2121 | 2130 | 2136 | 2136 | 2112 |
| | C$_{ant}$ ($\mu$mol kg$^{-1}$) | 33 | 36 | 27 | 20 | 22 | 40 |
| 1991-1997 | Thickness (m) | 244 | 163 | 1197 | 186 | 572 | 196 |
| | DIC ($\mu$mol kg$^{-1}$) | 2149 | 2151 | 2151 | 2149 | 2152 | 2144 |
| | DIC$_{nat}$ ($\mu$mol kg$^{-1}$) | 2121 | 2123 | 2127 | 2135 | 2136 | 2113 |
| | C$_{ant}$ ($\mu$mol kg$^{-1}$) | 28 | 29 | 24 | 14 | 17 | 31 |
| 2000-2007 | Thickness (m) | 306 | 211 | 858 | 182 | 450 | 551 |
| | DIC ($\mu$mol kg$^{-1}$) | 2161 | 2159 | 2158 | 2156 | 2158 | 2153 |
| | DIC$_{nat}$ ($\mu$mol kg$^{-1}$) | 2129 | 2123 | 2131 | 2135 | 2136 | 2112 |
| | C$_{ant}$ ($\mu$mol kg$^{-1}$) | 32 | 36 | 27 | 20 | 22 | 41 |
| 2008-2015 | Thickness (m) | 361 | 705 | 527 | 136 | 413 | 416 |
| | DIC ($\mu$mol kg$^{-1}$) | 2162 | 2161 | 2161 | 2161 | 2161 | 2158 |
| | DIC$_{nat}$ ($\mu$mol kg$^{-1}$) | 2121 | 2117 | 2131 | 2138 | 2136 | 2112 |
| | C$_{ant}$ ($\mu$mol kg$^{-1}$) | 41 | 44 | 30 | 24 | 25 | 46 |