# Peer review of "Inorganic Carbon and Water Masses in the Irminger Sea since 1991"

_Biogeosciences, 2017_

## Referee Comment (RC1) · Anonymous Referee #1 · 9 Mar 2017

I applaud the authors for a thorough treatment of a large series of hydrographic and carbon data collected over more than two decades. This is a nice contribution to the understanding of the marine carbon cycle in a changing climate in a region where substantial ventilation of the oceans occurs. In general the manuscript is well organized and easy to read, with one exception; there are too many abbreviations used. I have no problems with using abbreviations for water masses as well as currents, but don't see the point of doing this for regions like the "subpolar North Atlantic" or the "subpolar gyre". It cost very little to spell these out. Additionally I have some minor comments that the authors can consider before this manuscript is suitable for publication in Biogeosciences.

P 2, L 32. Delete last word, "conditions".

P 5, L20 -. You apply the data from 100-200 m to determine the disequilibrium of DIC as well as preformed TA. Some of the waters within the water column left the surface at very different regions, e.g. the DSOW. How well does the "observed" parameters represent these waters and does this aspect impact the computations? Some words on this would be nice.

P 5, L 30. SWT needs to be identified in the text, not only in Table 2.

P 7, L 7. Ideally all hydrographic parameters should be independent if one wants the best quality of a eOMP analysis, and this would be worth noting. That is a further reason why it is wise to limit the number of water masses in each analysis, as is done.

P 7, L 27. Here the SWT concentrations of the different parameters are referred to, thus relying on the disequilibrium of DIC and preformed TA as mentioned above. It would be nice for the reader to have these concentrations specified for each SWT.

P8, L6. It reads "multiplied by the bottom depth". However, the computations are performed in a set of density layers, see P 7, L 11 -, so why by bottom depth. Please expand text to address this.

P 9, L 25 and continuation. There are two reasons why DIC-nat can change, either a change in salinity (i.e. mixing or other water mass) or in primary production / remineralization of organic matter. The salinity effect is well discussed in respect to the layer thickness aspect. But the mineralization aspect is not much discussed. It would be nice to see some discussions of this in relation to the convection, e.g. on L 30 same page. The newly ventilated water has higher DIC-anthro but the water that is expelled likely had more remineralized DIC. Similar aspects relate to the text on P 11, paragraph starting L 15. Here it reads that all SWTs except cLSW has a constant DIC-nat, while the cLSW gets accumulation of DIC during ageing, presumably be remineralization. That such an increase is not seen in the other SWTs is that explained by a steady state situation, or? What is then the difference for the cLSW. Is there a difference between the SWTs or is this a result of the uncertainty of the method? Some discussions

on this would be valuable, more than now on P 13, L 5.

P 11, L 25. This sentence needs some rewriting. I can guess what is meant, but "The decomposition of the inventory rate of change. . ." is not very informative.

---

## Referee Comment (RC2) · Anonymous Referee #2 · 19 Jul 2017

General comments:

The present study is developed in a very important oceanic region in terms of the carbon system. The results obtained are based on high quality measurements and demonstrate the importance of maintaining the effort of the international community in carrying on high-quality measurements on world-wide repeat lines. In general, the manuscript is well written and the results well presented. Besides, the results add insight into the changes in DIC by analysing the changes in its natural and anthropogenic components. Nevertheless, there are some comments on the manuscript that need to be clarified before the manuscript can be accepted for publication in Biogeosciences.

Major comments:

[Figure]

I do not understand very well the use of the thickness of the water mass layers as a driving mechanism to the changes in the carbon storage rates. As far as I know, the authors are not exactly computing the thickness of a layer. The computation of the layer thickness (Eq. 5, page 8) implies that there are points that "share thickness" between SWTs, isn't it? I understand that the authors are computing how thick in the water column the distribution of the SWTs are. I would like the authors to clarify this in the text. Thus, the dependence/not-dependence of on the storage rates of natural and anthropogenic DIC on the SWT-layer thickness is intrinsic to the computation of the layer thickness and the use of the OMP to estimate the anthropogenic DIC. I do not think that the decomposition into layer thickness driven changes and concentration driven changes (Figures 9 and 10) is really needed. The results of the OMP should not be used as a driving mechanism. The increase/decrease of the layer thickness is the solution of the OMP to the mixing between the water masses. The authors found the layer thickness a driving mechanism because the OMP results are used to establish the amount of anthropogenic carbon in the interior ocean.

Minor comments:

Abstract: Page 1, line 6. It should be mention that the distribution of the main water masses is based on the results from an OMP analysis.

4.1 Anthropogenic CO2 calculation: Page 5. The authors should mention how AT0 (preformed alkalinity?) and $\Delta$Cdis are estimated in the interior ocean, i.e., using the OMP (Vazquez-Rodriguez et al., 2009).

5 Results: Page 9, lines 30-32. I am not sure about the comparison between the increase in the Cant inventory from 2012 to 2015, which I also think could be real, and the peak in 2005. Between 2012 and 2015 there are only two cruises that were measured in summer and spring, respectively, when the active convection is not as deep as in winter. Nevertheless, the peak in 2005 pops up in the group of cruises of 2004, 2005 and 2006 and the cruise of 2005 was measured in October, closer to winter

and more likely to have some episode of active convection (even though a general situation was a neutral NAO state). Could the peak of 2005 be due to interannual variability?

5.3 DIC storage rate decomposition: Pages 11-12. See major comments.

6 Discussion: Page 13, line 21. pm should be $\pm$.

6 Discussion: Page 14, lines 11-27. These two paragraphs do not need Figure 10 for the discussion of the results. Graphs similar to Figure 6 would be better. It could also be interesting and complete more the discussion to relate the changes in oxygen, AOU to the changes in natural DIC. Some hints of it had been said in 5.2 section, page 11, lines 22-23 but not enough.

7 Conclusions: Page 15, lines 5-8. See major comments.

---

## Author Comment (AC1) · 22 Sep 2017

To the two anonymous reviewers,

We would like to thank both reviewers for the careful consideration of our manuscript. We have revised our manuscript taking your constructive assessment into account. Please find below the original comments in black and our responses in blue; the revised manuscript sections are indicated in italics.

In general, we would like to clarify that the eOMP that we set up for the water mass analysis in the Irminger Sea between 1991 and 2015 is only used to determine the fraction of source water types in our region of interest and it is independent of the eOMP analysis used within the $\varphi C_T°$-method to calculate anthropogenic carbon. This seems to has been a source of misunderstanding and has been clarified in the revised manuscript.

**Reviewer 1**

I applaud the authors for a thorough treatment of a large series of hydrographic and carbon data collected over more than two decades. This is a nice contribution to the understanding of the marine carbon cycle in a changing climate in a region where substantial ventilation of the oceans occurs. In general the manuscript is well organized and easy to read, with one exception; there are too many abbreviations used. I have no problems with using abbreviations for water masses as well as currents, but don't see the point of doing this for regions like the "subpolar North Atlantic" or the "subpolar gyre". It cost very little to spell these out. Additionally I have some minor comments that the authors can consider before this manuscript is suitable for publication in Biogeosciences.

> We agree fully with the reviewer. We have taken out the abbreviations for 'subpolar gyre', 'subpolar North Atlantic', 'Irminger Current' and 'East Greenland Current' accordingly.

P 2, L 32. Delete last word, "conditions".

> Done.

P 5, L20 -. You apply the data from 100-200 m to determine the disequilibrium of DIC as well as preformed TA. Some of the waters within the water column left the surface at very different regions, e.g. the DSOW. How well does the "observed" parameters represent these waters and does this aspect impact the computations? Some words on this would be nice.

> We would like to clarify that the disequilibrium of DIC and preformed TA for waters colder than 5°C are, in the $\varphi C_T°$ method, determined using an eOMP analysis that is independent from the eOMP that we use for our water mass analysis described in section 4.2. This $\varphi C_T°$-eOMP analysis determines for each sampling point the fractions of 6 source water types that dominate deep global ocean ventilation; each with an assigned disequilibrium of DIC (calculated from CFC values) and preformed TA value, which were determined from all data in the 100-200 m layer available for each source region. These assigned values are used in combination with the $\varphi C_T°$-eOMP determined water mass fractions to calculate

disequilibrium of DIC and preformed TA at each sampling point (Pérez et al., 2009, Pardo et al., 2011, 2012, Vazquez-Rodriguez et al. 2012). This approach thus, actually takes into account that for instance DSOW is formed in a very different region than that we analyze. All of this has been clarified in section 4 and 4.1 in the manuscript.

P 5, L 30. SWT needs to be identified in the text, not only in Table 2.

We thank the reviewer for that suggestion. We now list all SWTs used in our eOMP accordingly:

*"[...] property features in the Irminger Sea (Fig. 2) and that are used in the present analysis, namely Subarctic Intermediate Water (SAIW), Intermediate Water (IW), classical and upper Labrador Sea Water (cLSW and uLSW), Denmark Strait Overflow Water (DSOW), upper Northeast Atlantic Deep Water (uNEADW), Iceland-Scotland Overflow Water (ISOW), Icelandic Slope Water (IcSW), Irminger Sea Water (ISW), Subpolar Mode Water (SPMW) and North Atlantic Central Water (NACW)." (P6L29-32)*

P 7, L 7. Ideally all hydrographic parameters should be independent if one wants the best quality of a eOMP analysis, and this would be worth noting. That is a further reason why it is wise to limit the number of water masses in each analysis, as is done.

We thank the reviewer for that remark. Indeed, while linear independence of the parameters is assumed, it cannot be guaranteed for example for phosphate and nitrate, which may locally correlate and no additional information can be gained including both parameters. However, that might not be the case if source water masses of very different origin are mixed; then one might not loose that degree of freedom for the eOMP analysis. In our case, we circumvent this issue by limiting the number of source water types, in each of the 4 eOMP analyses that we perform, to a maximum number of 5, while using 7 hydrographic parameters and adding the mass residual.

We have added the following two statements in the manuscript accordingly:

*"[...], which is solved in a non-negative least square sense assuming that the parameters are linearly independent:" (P6 L10)*

*"[...] system of linear equations. However, $NO_3$ and $PO_4$ might be locally correlated, therefore one degree of freedom for the eOMP analysis is potentially lost. Therefore, the Irminger Sea was divided into four regions, defined such that each contained a maximum number of 5 SWTs to be determined with the 7 parameters plus mass conservation. This ensures over-determination of the system of mixing equations, which then can be solved in a non-negative least square sense. In the deep ocean [...]" (P7 L19-20)*

P 7, L 27. Here the SWT concentrations of the different parameters are referred to, thus relying on the disequilibrium of DIC and preformed TA as mentioned above. It would be nice for the reader to have these concentrations specified for each SWT.

We would again like to clarify that our Irminger Sea eOMP described in Sect 4.2 is not used to determine the concentrations of $C_{ant}$ (see response to comment on p. 5 l. 20, above). Our approach is to first calculate the $C_{ant}$ concentrations for each sampling point using the $\varphi C_T^\circ$ method. Then the eOMP is used to determine the fraction of each source water type in each sampling point. And then equation (4) is used (for $C_{ant}$) to determine the $C_{ant}$ concentration in each source water type. The reason for this approach is that while the values of the parameters used for our eOMP can be assumed to be time independent and they can be determined from the data in hand, this is not the case for $C_{ant}$. $C_{ant}$ is time dependent and cannot be directly determined for each SWT (in contrast to e.g. salinity); this we would need to do for each year, which is not possible with the data at hand.

The concept of archetypical concentrations is a way to resolve the unknown concentrations of any species in the SWTs, i.e. these concentrations can be calculated using the observed concentration of that species in the sampling points and the known source water fractions.

The disequilibrium of DIC and preformed TA for each SWT can of course be calculated for each SWT, similar to what has been done for $C_{ant}$ as described above, using equation (4). In that case, though, these values should be included in Table 3 (and not Table 2). Nevertheless, we have chosen not to do this as the variability in $C_{ant}$ among the SWTs is almost entirely be explained as a result of differences in their age, and the DIC disequilibrium and preformed alkalinity are not relevant.

In the revised version of the manuscript we convey more clearly that the Irminger Sea eOMP is not used for calculating $C_{ant}$, and explain more clearly that (and why) the $C_{ant}$ in each SWT is determined through Equation (4).

*"[...] using an extended Optimum MultiParameter analysis (eOMP, see Sect. 4.2). While the hydrographic parameters that describe a set of source water types (SWTs) used for the eOMP analysis are assumed to be time independent, the concentrations within each water mass of different species such as DIC or $C_{ant}$ vary over time and can therefore only be resolved by evoking the concept of water mass mixing averaged concentration, i.e. archetypal concentration (Álvarez-Salgado et al., 2013) (see Sect. 4.2). Finally, the inventory changes can be decomposed into contributions from changes in the archetypal concentration of the source water types and from changes in layer thickness of each water mass assuming linearity:"* P5L9-15

*"In order to determine SWT concentrations of time-varying species [...]" P8L14*

P8, L6. It reads "multiplied by the bottom depth". However, the computations are performed in a set of density layers, see P 7, L 11 -, so why by bottom depth. Please expand text to address this.

We would like to clarify that the calculations were preformed at sampling depths. The density was only used for delineating the SWTs potentially present at each

sampling depth. We have nevertheless revised this passage to better convey how the layer thicknesses were calculated, also in response to comments made by Reviewer 2:

*"Here, $\Sigma_j\, r_{ij} / n_k$ is the fraction of the total water column that each SWT occupies at each station, with $n_k$ describing the number of sampling points per station. The fraction is unitless and needs to be scaled to the total water column height, i.e. multiplied by the maximum depth of station, $d_{k,max}$. The average layer thickness over all stations of the Irminger Sea transect is the mean layer thickness. As each water parcel is a mixture of different water masses represented by $r_{ij}$, Eq. (5) allows to convert each composite to a measure of height." P5L17-19*

*"While the density boundaries were used to identify the set of potentially present SWTs, the eOMP analyses were performed at each sampling point." P7L35*

P 9, L 25 and continuation. There are two reasons why DIC-nat can change, either a change in salinity (i.e. mixing or other water mass) or in primary production / remineralization of organic matter. The salinity effect is well discussed in respect to the layer thickness aspect. But the mineralization aspect is not much discussed. It would be nice to see some discussions of this in relation to the convection, e.g. on L 30 same page. The newly ventilated water has higher DIC-anthro but the water that is expelled likely had more remineralized DIC. Similar aspects relate to the text on P 11, paragraph starting L 15. Here it reads that all SWTs except cLSW has a constant DIC-nat, while the cLSW gets accumulation of DIC during ageing, presumably be remineralization. That such an increase is not seen in the other SWTs is that explained by a steady state situation, or? What is then the difference for the cLSW. Is there a difference between the SWTs or is this a result of the uncertainty of the method? Some discussions on this would be valuable, more than now on P 13, L 5.

We agree with the reviewer that the discussion with respect to remineralization and ageing in the different water masses should be discussed in more depth.

Over the time period of 24-years, no significant trends in $DIC_{nat}$ were found for any SWT but cLSW. However, as illustrated in Figure 8b, on shorter timescales the $DIC_{nat}$ concentration of individual SWTs shows quite some variability, particularly for the upper SWTs. For IW and uLSW the convection events in 2007/8, 2011/12 and 2014/15 can be depicted in the time series of $DIC_{nat}$: the renewal of these water masses due to deep(er) convection resets the water column in terms of remineralization. It is also worth noting that the system is not a closed one, which is why the observed concentration changes reflect as well different residence times of one water mass in the Irminger Sea, which would contribute to the (non-)significant concentration changes in different water masses.

We clarified this by adding the following statements:

*"[...] inventory varies slightly more. $DIC_{nat}$ is increasing as water masses age and $DIC_{nat}$ from remineralisation of organic matter accumulate, while it decreases during water mass renewal/ventilation, which brings water with preformed, relatively low, $DIC_{nat}$ concentrations into the ocean interior. It is notable that [...]" P10L13-15*

*"[...] Irminger Sea (Fröb et al., 2016). During the strong convection, older water masses enriched in $DIC_{nat}$ were replaced by water masses high in $C_{ant}$ due to their most recent contact to the atmosphere, and relatively low in $DIC_{nat}$ as $DIC_{nat}$ from remineralisation has not yet accumulated. In contrast to that, [...]" P10L18-20*

P 11, L 25. This sentence needs some rewriting. I can guess what is meant, but "The decomposition of the inventory rate of change ..." is not very informative.

Done. It now reads:

*"The decomposition of the inventory changes reveals the contribution that changes in the SWT distribution and changes in the concentration within these SWTs have on the total storage rate of DIC and its natural and anthropogenic components."*

**Reviewer 2**

General comments:
The present study is developed in a very important oceanic region in terms of the carbon system. The results obtained are based on high quality measurements and demonstrate the importance of maintaining the effort of the international community in carrying on high-quality measurements on world-wide repeat lines. In general, the manuscript is well written and the results well presented. Besides, the results add insight into the changes in DIC by analysing the changes in its natural and anthropogenic components. Nevertheless, there are some comments on the manuscript that need to be clarified before the manuscript can be accepted for publication in Biogeosciences.

Major comments:
I do not understand very well the use of the thickness of the water mass layers as a driving mechanism to the changes in the carbon storage rates. As far as I know, the authors are not exactly computing the thickness of a layer. The computation of the layer thickness (Eq. 5, page 8) implies that there are points that "share thickness" between SWTs, isn't it? I understand that the authors are computing how thick in the water column the distribution of the SWTs are. I would like the authors to clarify this in the text.

We thank the reviewer for this comment. Yes, the term thickness does not refer to strict density layers, it is an alternative, depth-normalized approach of expressing fractions of SWTs.

We have now clarified in the text the term thickness, also in response to comments made by reviewer 1 (px lY):

*"Here, $\Sigma_j\, r_{ij} / n_k$ is the fraction of the total water column that each SWT occupies at each station, with $n_k$ describing the number of sampling points per station. The fraction is unitless and needs to be scaled to the total water column height, i.e. multiplied by the maximum depth of station, $d_{k,max}$. The average layer thickness over all stations of the Irminger Sea transect is the mean layer thickness. As each water*

Thus, the dependence/not-dependence of on the storage rates of natural and anthropogenic DIC on the SWT-layer thickness is intrinsic to the computation of the layer thickness and the use of the OMP to estimate the anthropogenic DIC. I do not think that the decomposition into layer thickness driven changes and concentration driven changes (Figures 9 and 10) is really needed. The results of the OMP should not be used as a driving mechanism. The increase/decrease of the layer thickness is the solution of the OMP to the mixing between the water masses. The authors found the layer thickness a driving mechanism because the OMP results are used to establish the amount of anthropogenic carbon in the interior ocean.

> Here we would like to clarify that the eOMP that we used to calculate SWT fractions (Sect 4.2) is different from the eOMP used in the calculations of the $C_{ant}$ concentrations. $C_{ant}$ was calculated using the $\varphi C_T^\circ$ method, which uses an eOMP to estimate preformed alkalinity and the disequilibrium of DIC below the 5°C isotherm, which are then used in the calculation of $C_{ant}$. This '$\varphi C_T^\circ$ eOMP' was set up for the global ocean and is based on 6 different water masses that dominate ventilation of the global ocean below the 5°C isotherm. Each of the 6 water types here have an assigned disequilibrium of DIC and preformed TA. The '$\varphi C_T^\circ$ eOMP' calculates the fraction of each of the source water types and thus the net disequilibrium of DIC and preformed TA at each sampling point, which are then used to determine $C_{ant}$. See Perez et al. (2008) and Vazquez-Rodriguez et al. (2009, 2012) for more detail.
>
> The eOMP used to calculate Irminger Sea SWT fractions (Sect. 4.2) is completely independent, giving a higher resolution of source water types that could ever be reached with an eOMP set up with only 6 water masses. And, importantly, it has not been used to calculate $C_{ant}$, which is thus independent of the layer thickness. Hence thickness and $C_{ant}$ are independent, and the decomposition is valid.

Minor comments:
Abstract: Page 1, line 6. It should be mention that the distribution of the main water masses is based on the results from an OMP analysis.

> Yes, we agree and revised the mentioned statement accordingly:
>
> *"Here, data from 15 cruises in the Irminger Sea covering the 24-year period between 1991 and 2015 were used to determine changes in total DIC and its natural and anthropogenic components. Based on the results of an extended Optimum Multiparameter Analysis (eOMP), the inventory changes are discussed in relation to the distribution and evolution of the main water masses." P1L6-7*

4.1 Anthropogenic CO2 calculation: Page 5. The authors should mention how AT0 (preformed alkalinity?) and ΔCdis are estimated in the interior ocean, i.e., using the OMP (Vazquez-Rodriguez et al., 2009).

Yes, we agree with the reviewer and clarified the passage in section 4.1 (see below). As stated earlier, the eOMP method used to determine preformed alkalinity and the disequilibrium term below the 5°C isotherm has been set up by Vázquez-Rodríguez et al. (2009), and is different to the eOMP method described in section 4.2. The '$\varphi C_T°$ eOMP' is based on 6 SWTs types that dominate deep global ocean ventilation. As Vázquez-Rodríguez et al. (2009) show, the $\varphi C_T°$ estimates for anthropogenic carbon are realistic in the North Atlantic.

We added the following to section 4.1:

*"[...] and the air-sea $CO_2$ disequilibrium ($\Delta C_{dis}$). The approach involves the following basic features: The subsurface layer (100-200 m) preserves conditions during water mass formation and is therefore taken as a reference. $\Delta C_{dis}$ is parameterized based on subsurface data using a short-cut approach to calculate $C_{ant}$. The set of parameterizations for preformed alkalinity ($A°_T$) and $\Delta C_{dis}$ obtained from the subsurface data are applied directly to waters with temperatures larger than 5°C. For waters below the 5°C isotherm, an extended Optimum MultiParameter (eOMP) analysis was used to estimate $A°_T$ and $\Delta C_{dis}$, which was successfully used in previous studies (Pérez et al., 2008; Vázquez-Rodríguez et al., 2009, 2012). This eOMP determines in each sampling point the fraction of 6 water masses that ventilate the global ocean, taking different formation histories and water mass origins into account. Each water mass has assigned values for $A°_T$ and $\Delta C_{dis}$ and together with the obtained fractions, $A°_T$ and $\Delta C_{dis}$ can be calculated.*

*The major advantage of the $\varphi C_T°$ method over other back-calculation methods is that is does not rely on measurements of age tracers, such as chlorofluorocarbons (CFCs). Further, the parameterized $A°_T$ is corrected for effects of $CaCO_3$ dissolution changes and the sea surface temperature increase since preindustrial times and any spatial and temporal variability of $\Delta C_{dis}$ is taken into account. [...]"*

5 Results: Page 9, lines 30-32. I am not sure about the comparison between the increase in the Cant inventory from 2012 to 2015, which I also think could be real, and the peak in 2005. Between 2012 and 2015 there are only two cruises that were measured in summer and spring, respectively, when the active convection is not as deep as in winter. Nevertheless, the peak in 2005 pops up in the group of cruises of 2004, 2005 and 2006 and the cruise of 2005 was measured in October, closer to winter and more likely to have some episode of active convection (even though a general situation was a neutral NAO state). Could the peak of 2005 be due to interannual variability?

We agree with the reviewer that the statement in question needs clarification. It is unlikely that the peak in $C_{ant}$ in 2005 is related to deep convection that same winter. October is still at the onset of the convective season and the mixed layer depth has generally not deepened enough to generate a signal that strong (Våge et al., 2008). On the other hand, what could be possible is that convectively formed water with a higher $C_{ant}$ signature has advected into the Irminger Sea. Labrador Sea Water formed via convection in Labrador Sea takes approximately 1,5-2 years to travel into the Irminger Sea; water formed south of Cape Farewell needs less time (Straneo et al., 2003, Palter et al., 2015). The peak in uLSW thickness in 2005 can be explained by such a mechanism, and there was moderately strong deep convection in the Labrador Sea in 2000-2003 (Yashayaev et al., 2017). An

argument against this hypothesis is that no such signals are evident in the mid-1990s, a period of time where much stronger convection in the Labrador Sea was observed.

While we know for certain that the signal in 2015 is true (as we were there on a cruise (Fröb et al., 2016)), we can not distinguish in this framework whether the 2005 signal in the data was generated locally or had been advected into the area, i.e. related to interannual variability, or is a result of measurement uncertainty. We have revised the section in question accordingly:

*"In contrast to that, the peak in 2005 in the $C_{ant}$ inventory could be related to the advection of $C_{ant}$ enriched water masses formed in the Labrador Sea or in the region south of Cape Farewell into the Irminger Sea (Straneo et al., 2003; Palter et al., 2016), but this strong signal may also reflect the true error or reveal measurement bias." P10L20-23*

5.3 DIC storage rate decomposition: Pages 11-12. See major comments.

We do still think that the decomposition increases our mechanistic understanding of the involved processes, and hope, the explanation above is sufficient.

6 Discussion: Page 13, line 21. pm should be ±.

Done.

6 Discussion: Page 14, lines 11-27. These two paragraphs do not need Figure 10 for the discussion of the results. Graphs similar to Figure 6 would be better. It could also be interesting and complete more the discussion to relate the changes in oxygen, AOU to the changes in natural DIC. Some hints of it had been said in 5.2 section, page 11, lines 22-23 but not enough.

We still think that Figure 10 still is a useful representation as it shows that particularly on short timescales the abiotic DIC term can not be used as a representation of $C_{ant}$. We have added the following statements:

*"[...] despite stronger convection in the early 1990s. At the same time, the small increase in AOU does neither reflect the constant inventory in $DIC_{nat}$ before 1997, nor does the constant AOU reflect the loss in $DIC_{nat}$ after 1997. Again, this is attributed [...]" P15L18-19*

*"[...] convection in 2015 (Fröb et al., 2016). Here, the strong loss in AOU matches the loss in $DIC_{nat}$ as the water column is reventilated so that remineralized organic matter is replaced by a larger fraction of $C_{ant}$. In fact, [...]" P15L24-26*

7 Conclusions: Page 15, lines 5-8. See major comments

Same as above. In addition, we found that the layer thickness changes are more important for changes in the natural component, while it is in fact concentration changes due to air-sea gas exchange, ventilation and/or remineralization that drive changes in DIC and $C_{ant}$.

---

## Author Comment (AC2) · 22 Sep 2017

The comment was uploaded in the form of a supplement:
https://www.biogeosciences-discuss.net/bg-2017-27/bg-2017-27-AC2-supplement.pdf